evolution, ecology

edaphic divergence, endemism, flowering time, phenological isolation, plasticity, serpentine

**Author for correspondence:**
Kathleen M. Kay
e-mail: kmkay@ucsc.edu

†Present address: Department of Plant and Microbial Biology, University of Minnesota - Twin Cities 1479 Gortner Avenue, St. Paul, Minnesota 55108, USA.

# Parallel evolution of phenological isolation across the speciation continuum in serpentine-adapted annual wildflowers

Shelley A. Sianta† and Kathleen M. Kay

Department of Ecology and Evolutionary Biology, University of California, 130 McAllister Way, Santa Cruz, CA 95060, USA

SAS, 0000-0003-1041-228X; KMK, 0000-0001-8858-110X

Understanding the relative importance of reproductive isolating mechanisms across the speciation continuum remains an outstanding challenge in evolutionary biology. Here, we examine a common isolating mechanism, reproductive phenology, between plant sister taxa at different stages of adaptive divergence to gain insight into its relative importance during speciation. We study 17 plant taxa that have independently adapted to inhospitable serpentine soils, and contrast each with a nonserpentine sister taxon to form pairs at either ecotypic or species-level divergence. We use greenhouse-based reciprocal transplants in field soils to quantify how often flowering time (FT) shifts accompany serpentine adaptation, when FT shifts evolve during speciation, and the genetic versus plastic basis of these shifts. We find that genetically based shifts in FT in serpentine-adapted taxa are pervasive regardless of the stage of divergence. Although plasticity increases FT shifts in five of the pairs, the degree of plasticity does not differ when comparing ecotypic versus species-level divergence. FT shifts generally led to significant, but incomplete, reproductive isolation that did not vary in strength by stage of divergence. Our work shows that adaptation to a novel habitat may predictably drive phenological isolation early in the speciation process.

## 1. Introduction

A major goal of speciation research is to understand the relative importance of different reproductive isolating mechanisms, both across taxa and at different time points during the speciation process [1,2]. Comparative studies that include taxa at different stages of speciation are a powerful way to understand general patterns of when reproductive barriers evolve during speciation [3–6]. Although intrinsic postzygotic isolation has been shown to accumulate gradually with time since divergence in several taxa [7–10], we understand less about the evolutionary tempo for ecologically mediated prezygotic barriers, which are often of paramount importance early in speciation [2,11,12]. Adaptive ecological divergence may lead to reproductive isolation evolving rapidly and/or unpredictably according to the particulars of any taxon or selective environment. Alternatively, it may be that similar selective pressures drive the parallel evolution of reproductive isolation in predictable ways [13–15]. Resolving this question requires examining the evolution of the same reproductive barrier across varying levels of genetic divergence in response to similar selective pressures.

The importance of a reproductive isolating mechanism in speciation may be influenced by the degree to which it is genetically based versus plastic. The idea that plasticity promotes phenotypic divergence and speciation—primarily through facilitating fast niche expansion and colonization early during speciation—has a long history [16–18]. However, plastic reproductive isolation can break down with dispersal or environmental change, and taxa

with more genetically based barriers are more likely to proceed towards speciation [19,20]. Moreover, genetic barriers may be associated with speciation if taxa accumulate genetic differences contributing to reproductive isolation over time. Thus, genetically based barriers may both promote, and result from, speciation. Additionally, selection can make initially plastic reproductive isolation permanent through canalization if the plasticity is adaptive or reduce it through countergradient selection if the plasticity is simply a maladaptive stress-mediated response to a marginal habitat [21]. These multiple processes all lead to the prediction that reproductive isolation will be more strongly genetically based than plastic for taxa further along the speciation continuum.

Phenological reproductive isolation due to differences in the timing of mating is often involved in ecologically driven speciation because mating cues can be intimately tied to environmental factors [22–24] and phenological shifts automatically increase assortative mating [25,26]. For example, in plants, the onset and duration of flowering time (FT) can both respond to divergent selection and cause phenological isolation (e.g. [26–28]). However, mating phenology can also be plastic [29,30]. Plasticity in FTs can cause populations in different habitats to experience reduced pollen flow. However, if seeds disperse between habitats and migrants survive to flower, migrants should have similar flowering schedules as local plants, reducing phenological isolation. While theory suggests plasticity in FTs can make early divergence more probable and faster [26,31], there is a dearth of empirical support as to whether or not plasticity in FTs promotes the long-term evolution of reproductive isolation on timescales important to speciation. Understanding the importance of phenological isolation at different stages in the speciation process requires understanding how often, at what stage, and to what degree genetically based versus plastic changes in phenology evolve following ecological divergence.

Serpentine soil-adapted plants present an opportunity to study the importance and evolution of phenological isolation following parallel ecological divergence. Serpentine soils are harsh, often rocky substrates, characterized by low Ca : Mg ratios, low nutrients, and high heavy metals, and they impose strong divergent selection across steep ecological gradients [32,33]. Adaptation to serpentine has occurred independently in at least 39 families within California [34], and has led to the evolution of species with populations on and off serpentine (tolerator species, *sensu* [35]) and species that only occur on serpentine (endemic species). Many serpentine endemic species have small geographical ranges relative to their sister species and are thought to arise through budding, or peripheral-isolate, speciation [35–37]. Moreover, shifts in FT are commonly noted in annual serpentine systems [30,32,33,38,39], with earlier flowering the most common pattern reported. Earlier flowering is hypothesized as a way to escape drought-inducing conditions of rocky serpentine soils. Alternatively, theory predicts that plants in stressful habitats should flower later because of resource constraints [40]. Phenological shifts are known at both the population and species level [32,41], indicating that phenological isolation may play an important role at different stages of the speciation process.

In this study, we examine FT divergence across 17 taxa pairs that represent independent adaptation to serpentine soil leading to either a serpentine tolerator or a serpentine endemic. We hypothesize that strong, genetically based FT shifts characterize serpentine endemics relative to their nonserpentine sister species, whereas weaker and more plastic FT shifts characterize serpentine populations within a tolerator species relative to nonserpentine populations of the same species. Each of our 17 pairs comprises a closely related serpentine and nonserpentine population, hereafter called sister taxa (figure 1a). Nine sister taxa pairs consist of a serpentine and nonserpentine population from within one tolerator species (tolerator sister taxa pairs). The other eight pairs comprise a serpentine population from an endemic species and a nonserpentine population from its sister species (endemic sister taxa pairs). We use a greenhouse-based reciprocal transplant experiment in field-collected soil to quantify genetic and plastic differences in both flowering onset—hereafter referred to as FT—and overall phenological isolation within each sister taxa pair (figure 1b–d).

Our main goals are to understand whether parallel ecological divergence results in parallel phenological isolation, and whether progress towards speciation relates to the degree to which FT divergence is genetically based versus plastic. We first ask whether FT shifts are common following adaptation to serpentine, and whether they relate to the degree of divergence in edaphic and/or climate factors [20,42]. We then compare the magnitude of FT divergence in tolerator versus endemic sister taxa pairs to understand when shifts in FT evolve in the speciation process. Third, we determine the degree to which FT divergence between sister taxa is plastic versus genetically based (figure 1c). Fourth, we use full FT distributions to calculate phenological isolation when sister taxa are in their home soils (mimicking the barrier to pollen flow) and when sister taxa are in a common soil environment (mimicking the barrier to gene flow following seed dispersal). We compare phenological isolation between these two gene flow contexts for each sister taxa pair to determine whether endemic sister taxa pairs have more permanent phenological isolation than tolerator sister taxa pairs (figure 1d). Lastly, we explore patterns of plasticity between sister taxa. Because evolutionary transitions from nonserpentine to serpentine soil are common [43], we treat the nonserpentine sister taxon as a proxy for the ancestral condition of serpentine soil-mediated plasticity in FT. We use phenotypic selection analyses in serpentine soil to ask whether plastic shifts in serpentine soil are adaptive or maladaptive, and to assess whether plasticity in serpentine taxa evolved following colonization of serpentine.

## 2. Methods

### (a) Study system and sister taxa pair selection

We chose nine annual tolerator sister taxa pairs and eight annual endemic sister taxa pairs (table 1, electronic supplementary material, table S1), each comprising one serpentine population and one nonserpentine population. We searched for spatially proximate sister taxa using CalFlora occurrence data to minimize environmental differences other than the edaphic habitat (see electronic supplementary material, appendix S1 for details). However, due to allopatric distributions our sister taxa pairs vary in their geographical distance (electronic supplementary material, table S1). Our final list of sister taxa pairs spans six plant families and nine genera.

We used internal transcribed spacer (ITS) ribosome sequences representing each taxon to quantify pairwise genetic divergence, as the number of nucleotide substitutions, as a

*Proc. R. Soc. B* **288**: 20203076

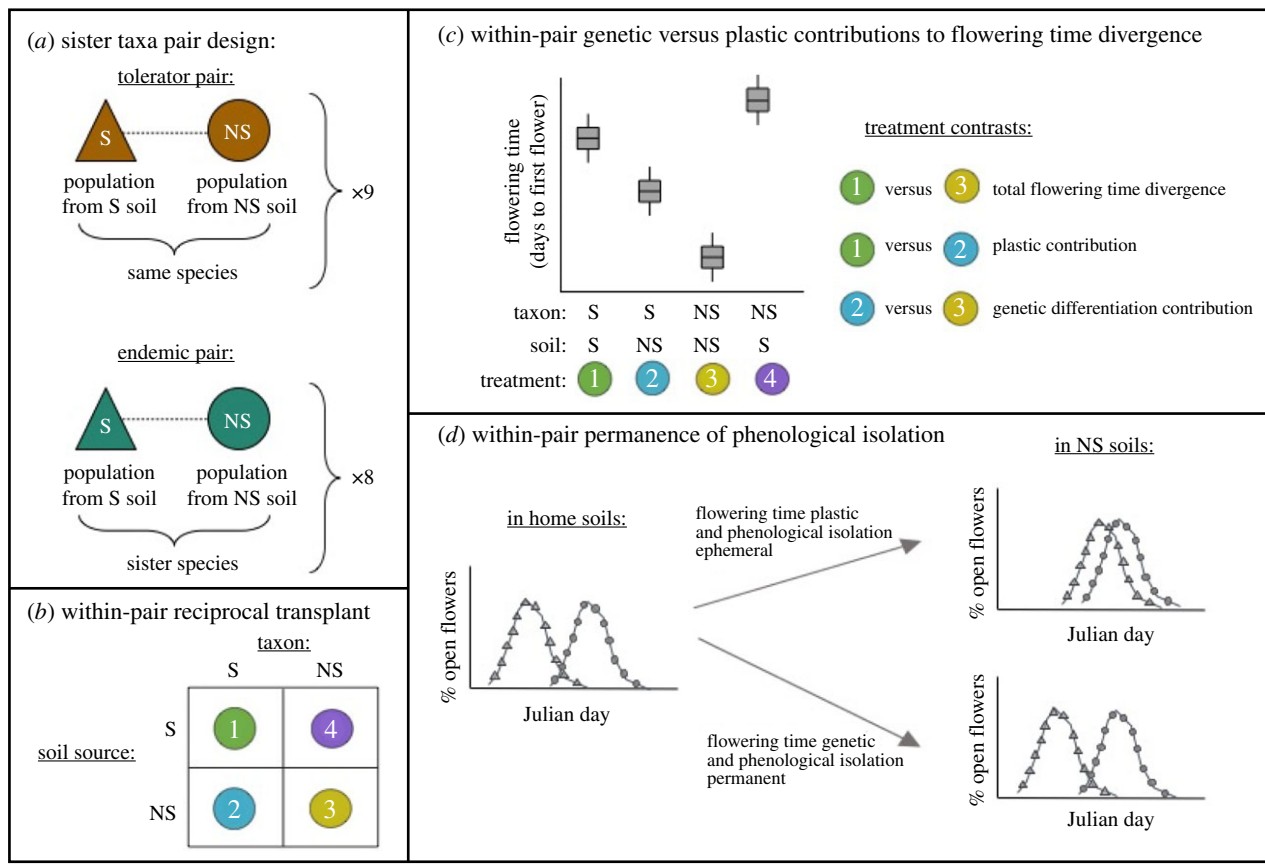

**Figure 1.** Conceptual diagram of the experimental design and data analysis. (*a*) The study is composed of nine within-species and eight between-species sister taxa pairs, each consisting of one serpentine (S) and one nonserpentine (NS) population. (*b*) For every pair, we reciprocally transplanted each taxon in field-collected soil. The four treatment numbers here are repeated in (*c*). (*c*) For every pair, we quantify total flowering time (FT) divergence (treatment contrast 1 and 3). Treatment contrasts 1 versus 2 and 2 versus 3 decompose the extent to which total FT divergence is plastic versus genetic, respectively. (*d*) For every pair, we quantify phenological isolation between sister taxa when each is grown in its home soil and when each is grown in a common NS soil. The difference between the two ecological contexts reflects the permanence of phenological isolation. Ultimately, within-pair metrics are used to ask whether endemic pairs have stronger, and more genetically based, FT divergence and phenological isolation than tolerator pairs.

**Table 1.** Sister taxa pairs used in this study. The serpentine (S) and nonserpentine (NS) taxon are from the same species in tolerator pairs and are from sister species in endemic pairs. Taxon codes are given for reference in subsequent figures. The NS taxon of endemic pairs come from either a tolerator (T) or non-tolerator (NT; no population occurrences on serpentine) species. Full details on sister taxa pairs are in electronic supplementary material, table S1.

| tolerator sister taxa pairs | endemic sister taxa pairs (S taxon; NS taxon) |
|---|---|
| *Clarkia concinna* (CACO) | *Navarretia jepsonii* (NAJP); *N. heterandra* (NAHN; NT) |
| *Clarkia breweri* (CABR) | *Navarretia rosulata* (NARS); *N. heterodoxa* (NAHX; T) |
| *Plantago erecta* (PLER) | *Clarkia gracilis* ssp. *tracyi* (CAGT); ssp. *albicaulis* (CAGA; T) |
| *Mimulus guttatus* (MGUT) | *Collomia diversifolia* (CLDV); *C. heterophylla* (CLHT; NT) |
| *Collinsia sparsiflora* (COSP) | *Layia discoidea* (LADI); *L. glandulosa* (LAGL; NT) |
| *Collinsia heterophylla* (COHT) | *Mimulus nudatus* (MNUD); *M. guttatus* (MGUT; T) |
| *Trifolium willdenovii* (TWILD) | *Collinsia greenei* (COGR); *C. sparsiflora* (COSP; T) |
| *Navarretia pubescens* (NAPB) | *Camissonia benitensis* (CABE); *C. strigulosa* (CAST; T) |
| *Navarretia heterodoxa* (NAHX) | |

proxy for time since divergence (electronic supplementary material, appendix S1). Endemic sister taxa pairs had on average more genetic divergence in ITS (mean = 4.62, standard deviation = 6.82) than tolerator sister taxa pairs (mean = 0.56, standard deviation = 1.33 nucleotide substitutions), supporting the taxonomic categories for our focal taxa. In all analyses, we incorporate the phylogenetic relatedness among pairs to account for the effects of shared ancestry on differences in trait divergence. We use ITS sequences from the serpentine taxon from

each pair to build the phylogeny among pairs, following methods outlined in [44].

### (b) Seed and soil collections

At each population, we collected seed from 30 to 40 maternal plants, henceforth families, haphazardly selected throughout the natural population. We avoided the collection of individuals within 1–2 m of each other to maximize genetic diversity.

Collected fruits were stored in coin envelopes at 4°C until planting. We collected approximately 15 l of soil from the top 20 cm from 5 to 6 locations within each population. We discarded any rocks that would not fit in the RayLeach Conetainers (3.8 × 21 cm) used in the greenhouse experiment, but otherwise retained natural variation in soil particle size. Soil from each population was homogenized before use in the experiment.

## (c) Greenhouse reciprocal transplant experiment

For each sister taxa pair, we set up a greenhouse reciprocal transplant experiment in field-collected soil (figure 1b). We sowed seed from 30 families of each population into each soil type, for a total of 120 plants per sister taxa pair, and 1950 individuals across all taxa pairs. Because of the large sample size of this experiment, we split the sister taxa pairs into two experimental rounds. In the first year (2016–2017; '2017') five tolerator pairs and three endemic pairs were grown and in the second year (2017–2018; '2018') four tolerator pairs and five endemic pairs were grown (electronic supplementary material, table S1). Specific growing conditions and timelines for the two experimental rounds, as well as differences between them, are detailed in electronic supplementary material, appendix S2. We include the year as a covariate in all analyses to account for differences in growing conditions across the two experimental rounds.

For all plants that survived to flower, we quantified FT as the number of days between germination and opening of the first flower. There was little to no variation among treatments within sister taxa pairs in germination timing as a function of our greenhouse conditions (electronic supplementary material, appendix S3), so we focus on differences in time-to-flower from the germination stage. Once individuals started flowering, we conducted weekly censuses of open flowers per plant. We did not hand-pollinate any of the plants in the greenhouse. Some of the taxa in the study readily self-pollinated, some underwent delayed self-pollination, and some did not set any self-seed, resulting in individual-flower lifespans and overall flower duration lasting longer in some species. However, mating system types were evenly spread among endemics and tolerators and only two pairs—one endemic (Collinsia greenei–C. sparsiflora) and one tolerator pair (C. sparsiflora–C. sparsiflora)—had sister taxa that varied in mating system.

## (d) Data analysis
### (i) Is FT divergence common following adaptation to serpentine, and is it correlated with greater edaphic and/or climatic divergence?

We characterized FT divergence within each sister taxa pair by quantifying flowering onset in each taxon when grown in its home soil, and testing for a difference between the two sister taxa using t-tests. We used sequential Bonferroni corrections to account for multiple comparisons.

Next, we determined if the absolute magnitude of FT divergence among pairs can be explained by multivariate divergence in (i) soil chemistry and texture, and (ii) climate (refer to electronic supplementary material, appendix S4 for full details). We used separate phylogenetic generalized least squares (PGLS) models, in which the phylogenetic variance-covariance matrix structures the error terms, for the soil and climate analyses. The absolute magnitude of FT divergence for each pair was modelled using a hierarchical Bayesian model (electronic supplementary material, appendix S5, Model 1), and we used the average value from the posterior distribution of mean FT divergence for each pair as the response variable in the PGLS models.

### (ii) Do endemic sister taxa pairs have greater FT divergence than tolerator sister taxa pairs?

We modelled absolute FT divergence within each sister taxa pair, and subsequently tested for differences between endemic and tolerator pairs while accounting for the phylogenetic relatedness among pairs using hierarchical Bayesian models (electronic supplementary material, appendix S5).

We modelled FT divergence among pairs as a function of a fixed intercept ($\beta_2$) that indicates the average magnitude of shifts in tolerator sister taxa pairs, a random intercept that accounts for phylogenetic relatedness among pairs ($\beta_0$), a fixed effect for pair type that indicates how different the average shift in endemic pairs is relative to tolerator pairs ($\beta_1$), and a fixed effect for year the pair was grown in the greenhouse ($\beta_3$). We were primarily interested in the effect of pair type on shifts in FT. If $\beta_1$ is greater than zero, endemic sister taxa pairs have greater FT divergence than tolerator sister taxa pairs. We quantified how much of the $\beta_1$ posterior distribution is greater than zero to assess the significance of the pair type effect. Because our model is a log-linear model (see electronic supplementary material, appendix S5), we exponentiate the coefficients to transform them into the units we are interested in.

### (iii) Is FT divergence more genetically based in endemic pairs than tolerator pairs?

Because serpentine taxa are primarily hypothesized to be derived from nonserpentine taxa, we examine the plasticity and genetic differentiation of each serpentine taxon relative to its possible nonserpentine progenitor (i.e. its nonserpentine sister). FT differentiation between members of a sister pair when each taxon is in its home soil comprises both genetic and plastic divergence (figure 1c, contrast 1 versus 3). However, in a pair's common nonserpentine, and putatively ancestral, soil, differences in FT between the sister taxa should be driven by genetic differentiation (and any maternal effects; figure 1c, contrast 2 versus 3). Conversely, differences in FT between the same serpentine taxon across the serpentine and nonserpentine soils primarily reflect plasticity (figure 1c, contrast 1 versus 2). We note that because we planted one sibling per maternal family into each treatment for a given taxon, differences in trait values among soil treatments is due to both plasticity and genetic differences between siblings.

For each sister taxa pair, we combined treatments 1, 2, and 3 (figure 1c) into a linear model with effects for taxon and soil, levelling our taxon and soil factors such that the nonserpentine taxon in nonserpentine soil treatment acted as the intercept. The taxon effect describes the change in FT between the two sister taxa in the nonserpentine soil—hereafter we refer to this effect as the 'genetic' effect. The soil effect describes the change in FT of the serpentine taxon between the two soil treatments—hereafter we refer to this effect as the 'plastic' effect. We calculated variance components for the genetic, plastic, and within-treatment effects by dividing the sum of squares for each source of variation by the sample size minus 1. We then used PGLS models to test whether the amount of non-residual variance in flowering onset contributed by plastic or genetic effects differs between endemic and tolerator sister taxa pairs.

### (iv) Do endemic sister taxa pairs have stronger and more permanent phenological isolation than tolerator sister taxa pairs?

We quantified phenological reproductive isolation, which captures divergence in both flowering onset and overall flowering duration, between each sister taxa pair, using a modified version of the Sobel & Chen [45] equation for phenological isolation (refer to electronic supplementary material, appendix S6 for

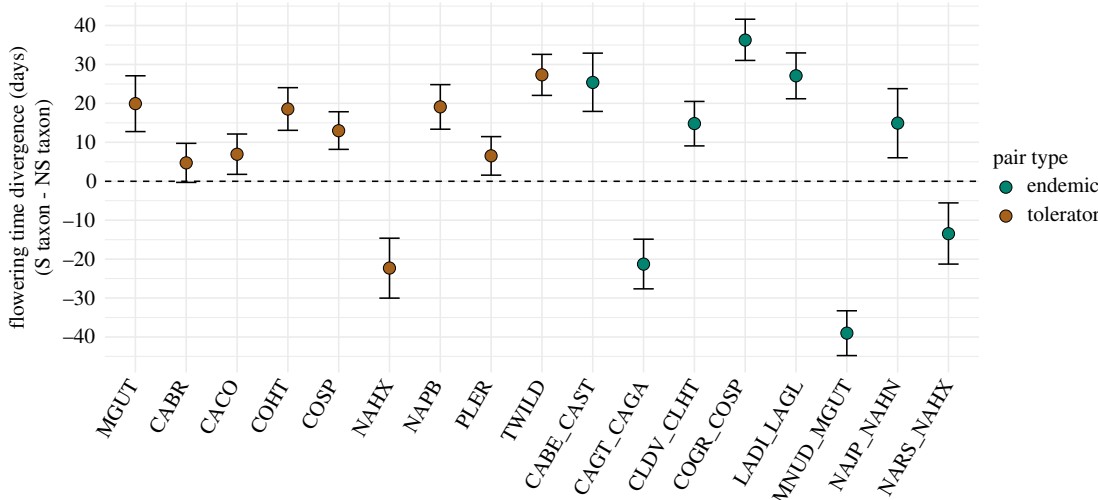

**Figure 2.** The majority of flowering time (FT) shifts in serpentine-adapted taxa are towards later flowering. Mean FT divergence when sister taxa were in their home soils was modelled with hierarchical Bayesian models. Points and error bars represent the mean and 95% credible intervals, respectively, of each pair's mean FT divergence posterior distribution. Positive values indicate the serpentine (S) taxon flowered later than the nonserpentine (NS) taxon and negative values indicate the S taxon flowered earlier than the NS taxon.

details). We used 10 000 bootstrap samples, resampling at the taxon and census day levels, to generate 95% confidence intervals around empirical measurements.

We used PGLS models to test for differences in phenological isolation between endemic and tolerator pairs in two ecological contexts (figure 1d): (i) when sister taxa are in their home soils and (ii) when sister taxa are in a common nonserpentine soil. We did not quantify phenological isolation in serpentine soils because most nonserpentine taxa used in this study had high mortality in serpentine soil, although we present FT distributions in serpentine soils in electronic supplementary material, figures S4–S6. We determined if phenological isolation is more permanent in endemic pairs than tolerator pairs by taking the difference of phenological isolation values obtained in these two ecological contexts for each taxa pair and using a PGLS model to test whether the difference is greater in endemic pairs than in tolerator pairs.

### (v) Has plasticity in FT evolved the following adaptation to serpentine and are plastic responses in an adaptive direction?

We compared siblings within maternal families grown in different soils to characterize serpentine-induced plasticity in FT. We calculated maternal family reaction norms by taking the difference in FT of an individual in nonserpentine soil with its sibling in serpentine soil—positive reaction norm values indicate later flowering in serpentine soils. We note that maternal family reaction norms include genetic differences between the two siblings, and with this caveat, use the term 'plasticity' to refer to the maternal family reaction norms. We assume that the nonserpentine taxon is a proxy for ancestral levels of FT plasticity, and that differences in FT plasticity between sister taxa are due to the evolution of plasticity in the serpentine taxon following adaptation to serpentine. Within each pair we tested for differences in the maternal family reaction norms with t-tests, and adjusted significance for multiple comparisons with a sequential Bonferroni correction. Five pairs were not included in the analysis because too few nonserpentine families survived in serpentine soil (electronic supplementary material, table S5).

To estimate whether plasticity in FT is adaptive, we quantified linear selection gradients on FT in serpentine soil by regressing relative total flower production on standardized flowering onset [46]. Because the sister taxa in some of the pairs had very different fecundities in serpentine soils, we estimated selection on each taxon separately. Taxa with fewer than 5 individuals

that survived to flower were excluded from the selection analysis (electronic supplementary material, table S6). We also estimated quadratic selection gradients (results in electronic supplementary material, table S2), but because only 1 taxon had significant quadratic effects after corrections for multiple comparisons, we focus on linear selection gradients below.

All analyses were run in R (v. 3.6.2). Refer to electronic supplementary material, appendix S7 for details on the packages used in each analysis.

## 3. Results

### (a) Is FT divergence common following adaptation to serpentine, and is it correlated with greater edaphic and/or climatic divergence?

Edaphic divergence is consistently associated with shifts in FT (the onset of flowering); 16 of the 17 sister taxa pairs had significant differences in FT when in their home soils (figure 2, electronic supplementary material, table S3), with an average shift of 19.4 days (s.e. 2.34 days) across all pairs. Absolute FT divergence ranged from 4.72 days in the *Clarkia breweri* tolerator pair to 39 days in the *Mimulus nudatus–M. guttatus* endemic pair. The serpentine taxon flowered significantly later than the nonserpentine taxon in 12 of the 16 pairs with an average shift of 18 days later (figure 2).

Multivariate soil distance explained 22.8% of the variation in absolute FT divergence (PGLS, $t_{17,14} = 2.03$, p-value = 0.061), such that sister taxa pairs with more divergent soil environments tended to have more divergent FT (electronic supplementary material, figure S1). We did not find any significant relationship between climatic distance and absolute FT divergence (PGLS $t_{17,14} = 0.52$, $p = 0.61$; electronic supplementary material, figure S2).

### (b) Do endemic sister taxa pairs have greater FT divergence than tolerator sister taxa pairs?

FT divergence was slightly larger in endemic pairs relative to tolerator pairs. The effect of pair type was marginally

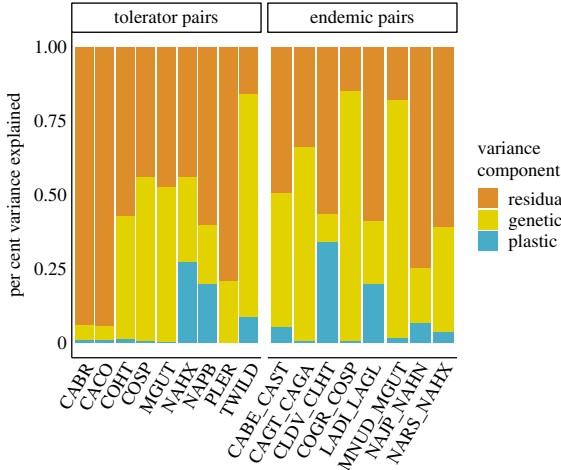

**Figure 3.** Characterization of the contribution of genetic versus plastic differentiation on total flowering time divergence between serpentine and nonserpentine taxa in their home soils.

significant, with 92.3% of the untransformed posterior distribution of $\beta_1$ greater than zero, indicating FT divergence is marginally greater in endemics. The untransformed posterior distribution of the $\beta_1$ coefficient had a mean of 0.46 and a 95% credible interval of (−0.17, 1.12). The average exponentiated $\beta_1$ coefficient was 1.58, which indicates FT divergence is greater in endemic pairs by 0.58 days (see electronic supplementary material, appendix S5 for details).

### (c) Is FT divergence more genetically based in endemic pairs than tolerator pairs?

Pairs varied in the proportion of plastic versus genetic contributions to total FT divergence (figure 3, electronic supplementary material, figure S3). All but two pairs showed a significant genetic effect, whereas only 5 of the 17 sister taxa pairs showed a significant plastic effect after corrections for multiple comparisons (electronic supplementary material, table S4). We found no difference between endemic and tolerator pairs in the relative contribution of plasticity to FT divergence (PGLS, $t_{17,14}$ (pair type) = −1.21, $p$-value (pair type) = 0.248, endemic mean (s.e.) = 0.22 (0.10), tolerator mean (s.e.) = 0.15 (0.07)), nor the relative contribution of genetic differentiation to FT divergence (PGLS, $t_{17,14}$ (pair type) = 1.21, $p$-value (pair type) = 0.248, endemic mean (s.e.) = 0.78 (0.10), tolerator mean (s.e.) = 0.84 (0.07)).

### (d) Do endemic sister taxa pairs have stronger and more permanent phenological isolation than tolerator sister taxa pairs?

There was no difference in phenological isolation between endemic and tolerator pairs when sister taxa were in their home soils (PGLS, $t_{17,14}$ = 0.48, $p$ = 0.68), nor when sister taxa were in the pair's nonserpentine soil (PGLS, $t_{17,14}$ = −0.62, $p$ = 0.54). In both ecological contexts, there was variation among sister taxa pairs in the strength of phenological isolation, ranging from 0.01 to 0.81 (in home soils: mean (s.e) = 0.32 (0.06); in nonserpentine soil: mean (s.e.) = 0.30 (0.05); electronic supplementary material, figures S4–S6).

We took the difference in phenological isolation values between the two ecological contexts to estimate the degree of

permanence of phenological isolation. We found a marginally significant effect of endemic pairs having more permanent phenological isolation than tolerator pairs, (electronic supplementary material, figure S7; PGLS, $t_{17,14}$ = 1.81, $p$ = 0.092). This pattern is driven by the three tolerator pairs, *N. heterodoxa*, *N. pubescens*, and *T. willdenovii*, that showed large decreases in phenological isolation when sister taxa were in a common nonserpentine soil compared to their home habitats.

### (e) Has plasticity in FT evolved following adaptation to serpentine and are plastic responses in an adaptive direction?

We compared the variation in maternal family reaction norms, our proxy for plasticity, between sister taxa to determine if plasticity in FT has evolved following edaphic divergence (electronic supplementary material, figure S8). There was low survival of the nonserpentine taxon in serpentine soil for many taxa pairs, which precluded $t$-tests for four sister taxa pairs (electronic supplementary material, table S5). Four of the 11 pairs tested, including three tolerator pairs and one endemic pair, showed a significant decrease in plasticity in the serpentine taxon compared to the nonserpentine taxon, the latter of which flowered later in serpentine soil (electronic supplementary material, table S5). However, low sample sizes of nonserpentine families that survived in both soils resulted in non-significant tests in pairs for which the serpentine taxon had qualitatively lower plasticity than the nonserpentine taxon (e.g. the *Collinsia heterophylla*, *C. sparsiflora*, and *N. jepsonii*–*N. heterandra* pairs).

We found overall support for selection for earlier flowering in serpentine soils at the taxon level among pairs (electronic supplementary material, figure S9 and table S6). Selection for earlier flowering was significant in 7 out of 17 of the serpentine taxa and 3 out of 11 of the nonserpentine taxa. However, after adjusting significance for multiple comparisons, selection for earlier flowering was significant in only four and one serpentine and nonserpentine taxa, respectively. There was no significant selection for later flowering in any taxa.

## 4. Discussion

Independent replicates of parallel ecological divergence provide insight into the evolution of ecologically driven reproductive isolation throughout the speciation process [47,48]. Here, we present the most comprehensive examination of phenological isolation across taxa that have experienced parallel selective environments. We used reciprocal transplant experiments to determine the extent of genetic and plastic shifts in FT in response to chemically and physically harsh serpentine soils. Surprisingly, we find that FT divergence is relatively predictable in plant lineages adapting to novel edaphic habitats. It is pervasive, largely genetic, and accumulates at the earliest stages of divergence, although it is rarely strong enough to cause complete reproductive isolation.

Phenological isolation has been championed as an important form of prezygotic isolation tightly linked to divergence in habitat use. Examples of its importance abound, in edaphic speciation of parapatric Lord Howe palms [49,50], recent plant colonization of toxic mine tailings [51], and young phytophagous insect species such as the apple maggot fly [52],

among many others. Much of this work has hinted at an important role for phenological isolation early in the speciation process, yet it has not been systematically investigated (but see [6] for an example within a single genus). Our findings of moderate phenological isolation both within- and between-species pairs support the importance of ecological divergence in driving early stages of plant speciation [53,54].

Instead of completing speciation, FT shifts may be more important in facilitating establishment and subsequent ecotypic divergence in novel habitats. Levin [49] argued that FT shifts promote niche expansion, because automatic assortative mating allows adaptation to newly colonized marginal habitats without the hindrance of maladaptive gene flow [25,55]. Because marginal habitats are often harsh relative to source population habitats, Levin argued FT shifts primarily would be stress-driven plastic responses. Here, we find that FT shifts in serpentine-adapted taxa relative to their nonserpentine sister taxa are primarily genetically based, even for populations within tolerator species (figure 3). Yet for those pairs that do show significant plasticity in FT in the serpentine taxon, serpentine-induced plasticity increases the magnitude of FT divergence. Thus, Levin's argument that plasticity increases FT divergence is supported in some of our pairs.

The majority of FT shifts were characterized by the serpentine taxon flowering *later* than the nonserpentine taxa. This pattern contrasts with the paradigm in serpentine and other rocky edaphic systems that adaptation to edaphic habitats with high water stress is accompanied by shifts to *earlier* flowering as a drought escape adaptation [30,33,56]. Higher water stress in serpentine soils could be due to faster water loss [39] or, when water content does not vary between serpentine and nonserpentine soils, the inability of serpentine plants to take up water efficiently [33]. Soil transplants inherently disrupt the soil structure and may change water relations [30,57], such that FT in the greenhouse does not fully reflect field FT. Nevertheless, field observations of coarse-scale FT distributions support the directionality of FT shifts for taxa pairs with marked shifts (electronic supplementary material, appendix S8). We also note that not all serpentine habitats are rocky, many nonserpentine habitats are rocky, and there is substantial variation in soil characteristics and water availability across soil types [32,44]. Thus, a mix of earlier and later flowering responses to serpentine soil is expected. Moreover, the serpentine soils used in this study have higher sand content than the nonserpentine soils [44], suggesting they do dry out more quickly. Nevertheless, shifts to later flowering reflect predictions of life-history theory, as plants living in more stressful habitats require more time to acquire necessary resources for reproduction [40].

That the majority of serpentine taxa flower later in serpentine soil despite evidence of selection for earlier flowering is a surprising result. Our plasticity measures suggest an intriguing possible explanation to the evolution of FT shifts following adaptation to serpentine in the context of this contradiction. Because serpentine taxa are often derived from nonserpentine taxa [43], it is possible that the serpentine-induced plasticity in the sister nonserpentine taxon represents an ancestral-like condition in each pair (as assumed in [58]). We found phenotypic selection on earlier flowering in serpentine soils in most taxa and thus hypothesize that plastic shifts experienced by nonserpentine taxa to extremely late flowering in serpentine soil are an example of maladaptive plasticity, presumably due to developmental constraints

in the nutrient-poor soil. We propose that as populations adapt to serpentine soils and evolve traits that increase nutrient uptake efficiency, selection drives the evolution of relatively earlier flowering, albeit still later than the nonserpentine taxon in its home soil. We see this pattern in 6 of the 17 sister taxa pairs (electronic supplementary material, figure S10). This proposed route to FT divergence is an example of countergradient selection leading to genetic compensation, wherein selection in the opposite direction of a plastic response leads to genetic change decreasing the amount of plasticity [21]. Countergradient selection in FT has been documented in other annual plants [59,60], including a study of ecotypes on and off of edaphically harsh mine tailings in *Thalspi caerulescens* [61].

We note that there was substantial variation among the sister taxa pairs in the magnitude of both FT divergence and overall phenological isolation. For example, FT shifts ranged from 4 to 40 days, and phenological isolation blocked 0–80% of potential gene flow between populations. Although our serpentine and nonserpentine source habitats vary considerably in soil chemistry and the competitive environment [44], the degree of habitat divergence only weakly predicted the magnitude of FT divergence. Interestingly, climatic differences between source habitats did not predict FT divergence, likely because many of our pairs are closely parapatric and share the same general climate. In addition, we coarsely categorized our taxa pairs into within- versus between-species pairs whereas they likely represent a continuous range of divergence, which may help explain the variation in FT shifts and phenological isolation. Moreover, this divergence may or may not represent stages of speciation; ecotypes within our tolerator species may never actually speciate, and endemics may have been driven to speciation by causes other than edaphic adaptation. Our results suggest that the presence of phenological isolation for taxa undergoing ecological divergence is relatively predictable, but the strength of that isolation is not.

Although our study is perhaps the most comprehensive examination of phenological isolation in response to a single axis of ecological divergence, the practicality of working in the greenhouse limits our inference. Our transplant studies captured the effects of soil chemistry and texture, without concomitant differences in soil structure, competitive environment, and water, all of which may alter plastic effects on FT in the field versus greenhouse. Our measures of phenological isolation may be sensitive to the lack of pollinators in the greenhouse because pollination and subsequent fruiting can strongly affect the duration of flowering, although endemics and tolerators were similarly affected by these factors. Nevertheless, we found clear, strong patterns that would not have been uncovered with a smaller set of field transplant studies. Despite these limitations, our data strongly support phenological isolation as a significant factor across stages of adaptive divergence.

## 5. Conclusion

Comparing the strength and degree of genetically based versus plastic differences in reproductive isolation across the speciation continuum reveals whether there are general patterns in how species evolve following adaptive divergence. We document the parallel evolution of genetically based FT divergence and moderate phenological isolation

following adaptation to serpentine soils. Our within- versus between-species comparisons demonstrate that phenological isolation evolves early in the speciation process and may be important for ecotypic formation. The role of plasticity in driving FT divergence is varied across taxa, but we find an intriguing pattern of countergradient selection, wherein colonization of serpentine soils causes maladaptive plastic shifts to later flowering, accompanied by selection for earlier flowering. Overall, our results show that genetic shifts in FT consistently accompany adaptive divergence in this system, and that plasticity interacts in nuanced ways to promote or constrain these shifts.

Ethics. Permissions required for collections were granted by: Marin Municipal Water District, permit no. R-15-07; County of San Mateo Parks Department, permit no. 20150803-ECP; Land Trust of Napa County Research Use Permit; US Fish and Wildlife Service, TE-163671.

Data accessibility. The raw data and scripts used to analyse data are available from the Dryad Digital Repository: https://doi.org/10.7291/D1RT0Q [62].

Authors' contributions. S.A.S. and K.M.K. designed the study. S.A.S. collected and analysed the data, and wrote the first draft of the manuscript. S.A.S. and K.M.K. both contributed to the final version.

Competing interests. We declare we have no competing interests.

Funding. S.A.S. was supported by a NSF Graduate Research Fellowship (NSF-DGE-1339067). The research was supported by the following fellowships and research grants awarded to S.A.S.: Rosemary Grant Award (Society for the Study of Evolution), Howard-Kohn Memorial Scholarship (California Native Plant Society, Marin Chapter), EEB Summer Research Fellowship (University of California, Santa Cruz), Research Scholarship (Garden Club of Naperville, IL).

Acknowledgements. We thank Jim Velzy, Molly Dillingham, and Sylvie Childress for help in the UCSC greenhouses. We thank the directors of the Donald and Sylvia McLaughlin Natural Reserve, Cathy Koehler and Paul Aigner, for their help in navigating the Reserve and providing support. Ryan O'Dell provided invaluable help getting around in the Clear Creek Management Area and collecting material.

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
