## [Peer Review File · Proceedings of the Royal Society B: Biological Sciences]

Review History

RSPB-2020-1298.R0 (Original submission)

Review form: Reviewer 1

Recommendation

Accept with minor revision (please list in comments)

Scientific importance: Is the manuscript an original and important contribution to its field?

Excellent

General interest: Is the paper of sufficient general interest?

Excellent

Quality of the paper: Is the overall quality of the paper suitable?

Good

Is the length of the paper justified?

Yes

Should the paper be seen by a specialist statistical reviewer?

No

Do you have any concerns about statistical analyses in this paper? If so, please specify them explicitly in your report.

No

It is a condition of publication that authors make their supporting data, code and materials available - either as supplementary material or hosted in an external repository. Please rate, if applicable, the supporting data on the following criteria.

Is it accessible?

Yes

Is it clear?

Yes

Is it adequate?

Yes

Do you have any ethical concerns with this paper?

No

Comments to the Author

This interesting paper describes a novel and significant experiment examining the role of genetic differentiation and phenotypic plasticity of flowering phenology in shaping reproductive isolation of sister taxa during colonization of serpentine soils. The strength of the study is the use of 17 serpentine/nonserpentine sister species pairs on native and non-native soils in a large greenhouse experiment. The authors find that genetically based phenological shifts and the degree of phenotypic plasticity both occur, but do not differ substantially between sister species pairs vs. within-species pairs of serpentine/nonserpentine ecotypes (contrary to their initial prediction that plasticity would be more important earlier in the process of taxon divergence). A surprising result from this study is that in the majority of taxa pairs serpentine taxa flower later than their non-serpentine sisters- even under common garden conditions. This finding is contrary to conventional wisdom that serpentine taxa flower earlier in the field and surprising in light of the phenotypic selection for earlier flowering on serpentine soil observed in this experiment.

In general, I am enthusiastic about this paper. The experiment is well designed, the results are interesting (albeit surprising), and it is well-written. However, I do have a few questions and comments.

Lines 101-103: This is conventional wisdom, but in how many species/populations has it actually been shown? Do the authors have field observations of approximate flowering phenology – or at least seed collection dates- for the taxa in this study? If so, do those observations show similar patterns to the results of the greenhouse experiment? Given the surprising finding of delayed flowering on serpentine, it would be good to show that this pattern is representative of field populations.

Lines 177-178. Is time from germination to flowering the most ecologically relevant metric? Did the seeds all germinate at the same time? Or were there differences within and among sister taxa pairs in days to germination? Were there differences in germination timing between soil types, which might have different hydrothermal properties? If so, then flowering date in Julian days would depend upon both germination and flowering phenology, and the reported metric might not tell the whole story. Hopefully this was not an issue, but the authors need to address this by explicitly discussing variation in germination phenology. It would also be useful to know a little more about the seasonal timeline of the experiment (e.g. watering timing) and how it aligned with the phenology of natural populations. Some of this is in the SI, but please point the reader to

it.

Line 194. An unfortunate typo!

Lines 228-237. I found the term “seed” effect to represent the genetic effect of taxon to be extremely confusing. I recommend finding a different terminology, e.g. “taxon” or “native soil” - if using the latter maybe refer to the “soil” effect as “soil treatment”.

Lines 278-283: Was there any evidence of non-linear or stabilizing/disruptive selection? Were there any covariates included? If there wasn't enough power for these tests, that's fine, but please say that. More importantly, why only measure selection on serpentine soil? Isn't the hypothesis that selection for early flowering will be stronger on serpentine than non-serpentine? If so, the test requires comparing selection gradients from both environments. I would also like to see a little more about the model used. Was the analysis done with absolute flower production (as implied by the text) or relative flower number per Lande and Arnold 1983?

Line 296- This sentence is not consistent with Figure 2 as I understand it. What happened to the negative shifts shown for some taxon pairs in that figure?

The idea of counter-gradient adaptation in phenology is interesting and plausible. But one interesting point that isn't really mentioned is that there's no evidence for selection for phenological reinforcement of reproductive isolation. That seems worth a mention.

Review form: Reviewer 2

Recommendation

Major revision is needed (please make suggestions in comments)

Scientific importance: Is the manuscript an original and important contribution to its field?

Acceptable

General interest: Is the paper of sufficient general interest?

Acceptable

Quality of the paper: Is the overall quality of the paper suitable?

Good

Is the length of the paper justified?

Yes

Should the paper be seen by a specialist statistical reviewer?

No

Do you have any concerns about statistical analyses in this paper? If so, please specify them explicitly in your report.

No

It is a condition of publication that authors make their supporting data, code and materials available - either as supplementary material or hosted in an external repository. Please rate, if applicable, the supporting data on the following criteria.

Is it accessible?

Yes

Is it clear?

Yes

Is it adequate?

Yes

Do you have any ethical concerns with this paper?

No

Comments to the Author

In this paper Siantis and Kay test whether repeated adaptation to serpentine soil both within and between species is associated with repeated genetic and plastic differences in phenology measured by flowering time (onset). The authors use species from across a broad taxonomic scope which makes the paper quite interesting. The authors used rigorous and appropriate statistical analyses in which they control for phylogeny to test whether the degree of flowering divergence is associated with evolutionary divergence and reproductive isolation. The authors find that divergence in phenology is a common occurrence when taxa are adapting to serpentine soil and that that divergence is primarily genetic rather than plastic. They find limited evidence for genetic compensation in flowering time.

I feel that while there are interesting results in this paper, it could be significantly improved in several ways. First of all the introduction and other parts of the text are somewhat confusingly written. In the introduction there is a lot of switching back and forth between talking about the roles of plastic vs genetic phenological isolation which makes it difficult to follow the exact big picture question that the authors are trying to answer. In general a lot of sections are wordy which makes them more difficult to follow and confusing, wordy names are used to identify treatments and effects. For example why use seed and soil effects when you mean genetic and plastic effects? See the specific comments below for more examples. Simplifying and clarifying the language used throughout the paper would make it more readable.

An experimental design issue that is not addressed in the manuscript is using maternal families rather than inbred lines to measure plasticity. This introduces a confounding effect since siblings are genetically different. This confounds genetic differences with environmentally induced plastic differences. The authors need to acknowledge this in the paper and justify their experimental design or interpret their results in light of this confounding factor.

Another possible experimental design issue not addressed is life-history variation. Since flowering time is often varies with other life-history traits it would be good to know whether the taxa used here differ in annual perennial life-history? Or are they all annuals? If they do differ (for example if most serpentine endemics are perennials), then this could have a confounding effect on flowering time differences in addition to serpentine soil and therefore it should be controlled for in the models.

In general the Figures are confusing. For example Figure 1 is much too text heavy. It makes it look busy and makes it hard to read. The authors should simplify each diagram and remove as much text as possible from the figure. They should use colors and shapes to indicate their meaning instead. For example in Figure 1D there is no need to have the soil icons next to each curve, you've already designated the curves as serp or non-serp based on circles and triangles. The icons add busyness. Figure 1C is so complex and confusing I really can't say what results it is communicating. The authors should try to dramatically simplify at least the first two figures. Specific suggestions are below.

Specific Comments:

Line 83: I'm getting a little confused by the introduction of plasticity here. Why are you switching back and forth between non-plastic and plastic changes in FT? If testing whether plasticity is important for RI is one of the goals of your study that idea should we introduced earlier and in a broader context.

Line 85-86: This transition right here is what is confusing about switching back and forth between plasticity and non-plastic FT divergence.

Lines 86-88: And where does plasticity fit into this goal? Please clarify.

Line 90: do you mean parallel ecological divergence? Similar is a little vague.

Lines 99-101: "Using natural variation in..." I find this phrasing confusing. Please rephrase for clarity.

Lines 108-109: "17 independent replicates of edaphic divergence associated with either ecotypic differentiation" This sentence is wordy and therefore confusing. I think the reader will assume that the edaphic divergence events you're studying are associated with phenotypic divergence. Otherwise why would you be talking about these taxa? So there doesn't seem to be a need to say that explicitly here.

Lines 122: I can see that "flowering onset" is a more precise description of flowering time, but as FT is the accepted term in the literature for the trait you are describing I would use that instead. Although the choice of language is ultimately up to the authors, it might make the paper easier to read.

Lines 123-124: What do you mean by habitat features? Be specific.

Lines 122-137: It might be helpful to start this paragraph with the big picture question you are addressing, and then move into all the specific subquestions. Otherwise the reader is likely to get a little lost in the specifics. Remind us what you're most excited about!

Line 134-135: Assuming this seems a bit presumptuous. Can you justify this assumption further?

Line 134-137: I am confused as to how you are going to do phenotypic selection specifically on plasticity in only one soil type...I of course read the rest of the paper and now do understand. But I have left this comment here so that you can see how readers might react to your description of your methods upon first reading this section. The description could be simplified/clarified.

Line 172-173: Could this have affected your measurements? How controlled are the greenhouses? Were there big environmental differences between the years?

Line 181: Typo. "We did not hand-pollinated.."

Line 191: It seems like a word like "difference" or divergence might be clearer here than shift.

Line 194: Typo. Hehe.

Line 197-199: Can you explain more about what these models control for? I'm assuming they ask how much divergence is due to environment after controlling for phylogenetic divergence, but I've never used them so would be useful to say something briefly about how they work here.

Line 225-227: "...reflect plasticity." Or they reflect within species/maternal family genetic variation. Or combination of within species genetic and env var. Did you plant genotypic

replicates such as inbred lines in each soil type or just species/maternal family replicates? Without true genetic replicates there is a genetic variation between treatments since even siblings are quite genetically different. You need to acknowledge this confounding factor of genetic variation as well as environmental variation in the paper.

Lines 231-233: This sentence is very confusing. It would be helpful to come up with simpler, less confusing names for these treatments than “serpentine seed serpentine” etc.

Lines 252-254: So does this mean that non-serp taxa were not grown in serp soil? That seems like a missed opportunity. Also only a partial reciprocal transplant...so using the term common garden seems more accurate.

Lines 257-259: Ah yes that makes sense. However still not quite a “reciprocal” transplant then.

Line 267-268: So this isn't a purely “plastic” response since siblings have genetic differences...

Lines 297-298: Interesting. Later is not what I would have expected.

Line 299: And soil distance is characterized by the ionic composition of the soil? By the particle size? By temperature and water measurements in the native habitats?

Lines 299-303: How does physical distance affect your flowering results? Was it controlled for in these analyses? There could be isolation by distance where pairs that are further apart from each other have undergone great neutral divergence in FT.

Line 316-319: WHY not just call these genetic and plastic effects? The seed and soil nomenclature is confusing and unnecessary.

Lines 330-33: Doesn't this imply that endemic pairs have a greater genetic divergence in flowering time than tolerator pairs? That seems to contradict what you found above in respect to plasticity. I'm confused how the two analyses are different...

Line 340: You need to acknowledge in the results that there may also be genetic differences between siblings in addition to plastic differences wrought by the environment.

Lines 346-349: This is an interesting trend though!

Lines 351-355: So the 7 out of 17 was before multiple testing correction? That is not quite clear from how you've presented thing here.

Lines 362-363: I feel that because your experiment was not truly “reciprocal” between each soil type you should call it a common garden, rather than a reciprocal transplant. That seems like the most accurate term.

Lines 366-367: Did you measure reproductive isolation due to flowering time? I don't remember reading about it in the results...

Lines 392-394: Yes that makes sense. Does it have anything to do with variation in life-history of the taxa? Were there perennials in your dataset? And if so do you control for the effect of life-history

Line 396-399: I am not sure I agree. Just because a closely related extant taxon has a characteristic does not mean that the most recent common ancestor of sister taxa shared that characteristic. It would be more conservative to say it is possible or probable that the plasticity is ancestral, but to assume it seems like a strong and not very scientific statement. I would rephrase.

Lines 413-415: Was this RI measurement in the results? I don't see it...maybe it was called something else?

Lines 424-426: Are there plans to do a follow up study in the field? And with inbred lines? That would be ideal!

Table 1: The genus's that have both a within and between species comparison seem particularly worthy of answering these questions. It might be worth highlighting their results separately if they're interesting.

Figure 1: In general this is a lot of text in this figure making it busy and complex looking. Find a way to show things visually with colors/shapes whenever possible rather than adding additional text and labels.

Figure 1 PartB: I would call this a common garden. Also it might be easier to visualize the setup in B if you had a space divided into quadrants like a punnet square instead of the current diagram. This diagram is a bit visually complex

Figure 1 Part C: Figure 1C is so complex and confusing I have no idea what data it is communicating. Please dramatically simplify.

Figure 1 Part D: Too busy!!! is no need to have the soil icons next to each curve, you've already designated the curves as serp or non-serp based on circles and triangles.

Figure 2: Again you don't need so much text. Indicate in a simpler Figure legend or in the caption that color indicates tolerator vs. endemic and the shape indicates soil type.

Figure 3: Nice. This is an easy to read figure.

Decision letter (RSPB-2020-1298.R0)

30-Jul-2020

Dear Dr Sianta,

I am writing to inform you that your manuscript RSPB-2020-1298 entitled "Parallel evolution of phenological isolation across the speciation spectrum in serpentine-adapted plants." has, in its current form, been rejected for publication in Proceedings B.

This action has been taken on the advice of referees, who have recommended that substantial revisions are necessary. With this in mind we would be happy to consider a resubmission, provided the comments of the referees are fully addressed. However please note that this is not a provisional acceptance.

Sincerely,
 Professor Loeske Kruuk
 mailto: proceedingsb@royalsociety.org

Associate Editor

Comments to Author:

We have now received two reviews of the paper entitled: "Parallel evolution of phenological isolation across the speciation spectrum in serpentine-adapted plants." Both reviewers and I all agree there is a very important contribution in this paper--seventeen species/population pairs and a very elegant design. The main issue, highlighted very well by reviewer 2 is that the main message of the paper is clear, but the connection between the conclusion and the results is very complicated and difficult for the reader. Figure 1 is a good start but it needs to be massively simplified and clarified such that the reader grasps 1) the hypotheses in the design, and 2) how the data will be used to test the different hypotheses. Reviewer 2 also highlights a number of possible improvements. I agree with all of reviewer 2's extensive and thoughtful comments, especially on the confusing presentation of Figure 1. I would add a few details: uncertainty is not conveyed very well in the figures--Figure 2 and 4 should have some uncertainty measure (e.g. SE) associated with each point. The authors should also check that their taxonomy is up-to-date. I believe the Jepson Herbarium has now moved the *Mimulus* species studied to *Erythranthe*.

Last, the countergradient result was a surprise when I got to it, since it was not mentioned in the abstract. The related question about whether serpentine soils truly are drier than nearby soils also deserves some attention. I'm not sure there is evidence to this effect. It's possible that serpentine soils are not actually drier, but that they "act" drier in late spring in some respects because of the relative growth rates of root systems on and off of serpentine, with the root systems on ultra-mafic soils slower to develop because of the overall slower rates of growth and carbon capture. Thus, by the time late spring occurs the root systems of non-serpentine individuals are set up to continue water uptake into May, while on the serpentine soils they are not large and extensive enough to continue. Field et al. (1997) and Hull and Wood (1984) might be interesting to look at from this perspective although this is not the main focus of either paper. Moreover, one of the points in the Field paper is that serpentine soils may in fact stay wetter, all else equal, because there is less leaf area to transpire water. All in all, it may be interesting to consider the countergradient results in light of this alternative hypothesis.

Field, Christopher, et al. "CO2 effects on the water budget of grassland microcosm communities." *Global Change Biology* 3.3 (1997): 197-206.

Hull, James C., and Sarah G. Wood. "Water relations of oak species on and adjacent to a Maryland serpentine soil." *American Midland Naturalist* (1984): 224-234.

Reviewer(s)' Comments to Author:

Referee: 1

Comments to the Author(s)

This interesting paper describes a novel and significant experiment examining the role of genetic differentiation and phenotypic plasticity of flowering phenology in shaping reproductive isolation of sister taxa during colonization of serpentine soils. The strength of the study is the use of 17 serpentine/nonserpentine sister species pairs on native and non-native soils in a large greenhouse experiment. The authors find that genetically based phenological shifts and the degree of phenotypic plasticity both occur, but do not differ substantially between sister species pairs vs. within-species pairs of serpentine/nonserpentine ecotypes (contrary to their initial prediction that plasticity would be more important earlier in the process of taxon divergence). A surprising result from this study is that in the majority of taxa pairs serpentine taxa flower later than their non-serpentine sisters- even under common garden conditions. This finding is contrary to conventional wisdom that serpentine taxa flower earlier in the field and surprising in light of the phenotypic selection for earlier flowering on serpentine soil observed in this experiment.

In general, I am enthusiastic about this paper. The experiment is well designed, the results are interesting (albeit surprising), and it is well-written. However, I do have a few questions and comments.

Lines 101-103: This is conventional wisdom, but in how many species/populations has it actually been shown? Do the authors have field observations of approximate flowering phenology – or at least seed collection dates- for the taxa in this study? If so, do those observations show similar patterns to the results of the greenhouse experiment? Given the surprising finding of delayed flowering on serpentine, it would be good to show that this pattern is representative of field populations.

Lines 177-178. Is time from germination to flowering the most ecologically relevant metric? Did the seeds all germinate at the same time? Or were there differences within and among sister taxa pairs in days to germination? Were there differences in germination timing between soil types, which might have different hydrothermal properties? If so, then flowering date in Julian days would depend upon both germination and flowering phenology, and the reported metric might not tell the whole story. Hopefully this was not an issue, but the authors need to address this by explicitly discussing variation in germination phenology. It would also be useful to know a little more about the seasonal timeline of the experiment (e.g. watering timing) and how it aligned with the phenology of natural populations. Some of this is in the SI, but please point the reader to it.

Line 194. An unfortunate typo!

Lines 228-237. I found the term “seed” effect to represent the genetic effect of taxon to be extremely confusing. I recommend finding a different terminology, e.g. “taxon” or “native soil”- if using the latter maybe refer to the “soil” effect as “soil treatment”.

Lines 278-283: Was there any evidence of non-linear or stabilizing/disruptive selection? Were there any covariates included? If there wasn't enough power for these tests, that's fine, but please say that. More importantly, why only measure selection on serpentine soil? Isn't the hypothesis that selection for early flowering will be stronger on serpentine than non-serpentine? If so, the test requires comparing selection gradients from both environments. I would also like to see a little more about the model used. Was the analysis done with absolute flower production (as implied by the text) or relative flower number per Lande and Arnold 1983?

Line 296- This sentence is not consistent with Figure 2 as I understand it. What happened to the negative shifts shown for some taxon pairs in that figure?

The idea of counter-gradient adaptation in phenology is interesting and plausible. But one interesting point that isn't really mentioned is that there's no evidence for selection for phenological reinforcement of reproductive isolation. That seems worth a mention.

Referee: 2

Comments to the Author(s)

In this paper Siantis and Kay test whether repeated adaptation to serpentine soil both within and between species is associated with repeated genetic and plastic differences in phenology measured by flowering time (onset). The authors use species from across a broad taxonomic scope which makes the paper quite interesting. The authors used rigorous and appropriate statistical analyses in which they control for phylogeny to test whether the degree of flowering divergence is associated with evolutionary divergence and reproductive isolation. The authors find that divergence in phenology is a common occurrence when taxa are adapting to serpentine soil and that that divergence is primarily genetic rather than plastic. They find limited evidence for genetic compensation in flowering time.

I feel that while there are interesting results in this paper, it could be significantly improved in several ways. First of all the introduction and other parts of the text are somewhat confusingly written. In the introduction there is a lot of switching back and forth between talking about the roles of plastic vs genetic phenological isolation which makes it difficult to follow the exact big picture question that the authors are trying to answer. In general a lot of sections are wordy which makes them more difficult to follow and confusing, wordy names are used to identify treatments and effects. For example why use seed and soil effects when you mean genetic and plastic effects? See the specific comments below for more examples. Simplifying and clarifying the language used throughout the paper would make it more readable.

An experimental design issue that is not addressed in the manuscript is using maternal families rather than inbred lines to measure plasticity. This introduces a confounding effect since siblings are genetically different. This confounds genetic differences with environmentally induced plastic differences. The authors need to acknowledge this in the paper and justify their experimental design or interpret their results in light of this confounding factor.

Another possible experimental design issue not addressed is life-history variation. Since flowering time is often varies with other life-history traits it would be good to know whether the taxa used here differ in annual perennial life-history? Or are they all annuals? If they do differ (for example if most serpentine endemics are perennials), then this could have a confounding effect on flowering time differences in addition to serpentine soil and therefore it should be controlled for in the models.

In general the Figures are confusing. For example Figure 1 is much too text heavy. It makes it look busy and makes it hard to read. The authors should simplify each diagram and remove as much text as possible from the figure. They should use colors and shapes to indicate their meaning instead. For example in Figure 1D there is no need to have the soil icons next to each curve, you've already designated the curves as serp or non-serp based on circles and triangles. The icons add busyness. Figure 1C is so complex and confusing I really can't say what results it is communicating. The authors should try to dramatically simplify at least the first two figures. Specific suggestions are below.

Specific Comments:

Line 83: I'm getting a little confused by the introduction of plasticity here. Why are you switching back and forth between non-plastic and plastic changes in FT? If testing whether plasticity is

important for RI is one of the goals of your study that idea should we introduced earlier and in a broader context.

Line 85-86: This transition right here is what is confusing about switching back and forth between plasticity and non-plastic FT divergence.

Lines 86-88: And where does plasticity fit into this goal? Please clarify.

Line 90: do you mean parallel ecological divergence? Similar is a little vague.

Lines 99-101: "Using natural variation in..." I find this phrasing confusing. Please rephrase for clarity.

Lines 108-109: "17 independent replicates of edaphic divergence associated with either ecotypic differentiation" This sentence is wordy and therefore confusing. I think the reader will assume that the edaphic divergence events you're studying are associated with phenotypic divergence. Otherwise why would you be talking about these taxa? So there doesn't seem to be a need to say that explicitly here.

Lines 122: I can see that "flowering onset" is a more precise description of flowering time, but as FT is the accepted term in the literature for the trait you are describing I would use that instead. Although the choice of language is ultimately up to the authors, it might make the paper easier to read.

Lines 123-124: What do you mean by habitat features? Be specific.

Lines 122-137: It might be helpful to start this paragraph with the big picture question you are addressing, and then move into all the specific subquestions. Otherwise the reader is likely to get a little lost in the specifics. Remind us what you're most excited about!

Line 134-135: Assuming this seems a bit presumptuous. Can you justify this assumption further?

Line 134-137: I am confused as to how you are going to do phenotypic selection specifically on plasticity in only one soil type...I of course read the rest of the paper and now do understand. But I have left this comment here so that you can see how readers might react to your description of your methods upon first reading this section. The description could be simplified/clarified.

Line 172-173: Could this have affected your measurements? How controlled are the greenhouses? Were there big environmental differences between the years?

Line 181: Typo. "We did not hand-pollinated.."

Line 191: It seems like a word like "difference" or divergence might be clearer here than shift.

Line 194: Typo. Hehe.

Line 197-199: Can you explain more about what these models control for? I'm assuming they ask how much divergence is due to environment after controlling for phylogenetic divergence, but I've never used them so would be useful to say something briefly about how they work here.

Line 225-227: "...reflect plasticity." Or they reflect within species/maternal family genetic variation. Or combination of within species genetic and env var. Did you plant genotypic replicates such as inbred lines in each soil type or just species/maternal family replicates? Without true genetic replicates there is a genetic variation between treatments since even siblings are quite genetically different. You need to acknowledge this confounding factor of genetic variation as well as environmental variation in the paper.

Lines 231-233: This sentence is very confusing. It would be helpful to come up with simpler, less confusing names for these treatments than “serpentine seed serpentine” etc.

Lines 252-254: So does this mean that non-serp taxa were not grown in serp soil? That seems like a missed opportunity. Also only a partial reciprocal transplant...so using the term common garden seems more accurate.

Lines 257-259: Ah yes that makes sense. However still not quite a “reciprocal” transplant then.

Line 267-268: So this isn't a purely “plastic” response since siblings have genetic differences...

Lines 297-298: Interesting. Later is not what I would have expected.

Line 299: And soil distance is characterized by the ionic composition of the soil? By the particle size? By temperature and water measurements in the native habitats?

Lines 299-303: How does physical distance affect your flowering results? Was it controlled for in these analyses? There could be isolation by distance where pairs that are further apart from each other have undergone great neutral divergence in FT.

Line 316-319: WHY not just call these genetic and plastic effects? The seed and soil nomenclature is confusing and unnecessary.

Lines 330-33: Doesn't this imply that endemic pairs have a greater genetic divergence in flowering time than tolerator pairs? That seems to contradict what you found above in respect to plasticity. I'm confused how the two analyses are different...

Line 340: You need to acknowledge in the results that there may also be genetic differences between siblings in addition to plastic differences wrought by the environment.

Lines 346-349: This is an interesting trend though!

Lines 351-355: So the 7 out of 17 was before multiple testing correction? That is not quite clear from how you've presented thing here.

Lines 362-363: I feel that because your experiment was not truly “reciprocal” between each soil type you should call it a common garden, rather than a reciprocal transplant. That seems like the most accurate term.

Lines 366-367: Did you measure reproductive isolation due to flowering time? I don't remember reading about it in the results...

Lines 392-394: Yes that makes sense. Does it have anything to do with variation in life-history of the taxa? Were there perennials in your dataset? And if so do you control for the effect of life-history

Line 396-399: I am not sure I agree. Just because a closely related extant taxon has a characteristic does not mean that the most recent common ancestor of sister taxa shared that characteristic. It would be more conservative to say it is possible or probable that the plasticity is ancestral, but to assume it seems like a strong and not very scientific statement. I would rephrase.

Lines 413-415: Was this RI measurement in the results? I don't see it...maybe it was called something else?

Lines 424-426: Are there plans to do a follow up study in the field? And with inbred lines? That would be ideal!

Table 1: The genus's that have both a within and between species comparison seem particularly worthy of answering these questions. It might be worth highlighting their results separately if they're interesting.

Figure 1: In general this is a lot of text in this figure making it busy and complex looking. Find a way to show things visually with colors/shapes whenever possible rather than adding additional text and labels.

Figure 1 PartB: I would call this a common garden. Also it might be easier to visualize the setup in B if you had a space divided into quadrants like a punnet square instead of the current diagram. This diagram is a bit visually complex

Figure 1 Part C: Figure 1C is so complex and confusing I have no idea what data it is communicating. Please dramatically simplify.

Figure 1 Part D: Too busy!!! is no need to have the soil icons next to each curve, you've already designated the curves as serp or non-serp based on circles and triangles.

Figure 2: Again you don't need so much text. Indicate in a simpler Figure legend or in the caption that color indicates tolerator vs. endemic and the shape indicates soil type.

Figure 3: Nice. This is an easy to read figure.

Author's Response to Decision Letter for (RSPB-2020-1298.R0)

See Appendix A.

RSPB-2020-3076.R0

Review form: Reviewer 3

Recommendation

Accept with minor revision (please list in comments)

Scientific importance: Is the manuscript an original and important contribution to its field?

Excellent

General interest: Is the paper of sufficient general interest?

Good

Quality of the paper: Is the overall quality of the paper suitable?

Good

Is the length of the paper justified?

Yes

Should the paper be seen by a specialist statistical reviewer?

No

Do you have any concerns about statistical analyses in this paper? If so, please specify them explicitly in your report.

No

It is a condition of publication that authors make their supporting data, code and materials available - either as supplementary material or hosted in an external repository. Please rate, if applicable, the supporting data on the following criteria.

Is it accessible?

No

Is it clear?

N/A

Is it adequate?

N/A

Do you have any ethical concerns with this paper?

No

Comments to the Author

I agree with the AE and previous reviewers that this manuscript is an important contribution to our understanding of repeated adaption and shifts in flowering time, especially because of the impressive number of pairs included in the study. The manuscript presents some interesting and unexpected results, and it was substantially improved during revision. For example, figure 1 is much clearer now, and uncertainly has been successfully added to the other figures. The authors also improved the manuscript by switching to "genetic/plastic" terminology and by setting up counter gradient selection in the introduction.

That said, I do have a few additional suggestions to further improve the manuscript's clarity (these are detailed below). Also, I think that the authors should more thoroughly respond to reviewer 1's comment about whether the later flowering is representative of field conditions in the main text. For example, they should mention that 11 of 12 previous studies found a shift to earlier flowering in the serpentine habitat. Finally, I'm not an expert on serpentine habitats, but I wonder whether greater competition limiting resource acquisition in the non-serpentine habitat could explain the discrepancy between the previous studies and this one.

Specific comments:

line 72-73: This prediction could be set-up more clearly. For example, that pattern could be caused because taxa with plastic barriers don't progress through the speciation continuum or because of canalization/counter gradient selection. It might be helpful to incorporate this entire paragraph into the following one that unpacks plasticity a little bit more.

line 113: I would point out the pairs that include tolerator species somewhere in the main text.

line 175: Why not include these results (and the results of the analysis that included germination timing) in the supplementary materials?

line 201: Specify that this is the absolute value

line 249: Is this also based on absolute values?

line 288-289: I would include language that points out that this result isn't very statistically robust here. E.g., "sister taxa pairs with more divergent soil environments tended to have more divergent FTs"

line 294: This is also a marginally significant effect.

line 320-322: It would be helpful to see the flowering time distributions for each pair in both soil types in the supplementary material.

Appendix line 81-83: Please include citations.

Review form: Reviewer 4

Recommendation

Major revision is needed (please make suggestions in comments)

Scientific importance: Is the manuscript an original and important contribution to its field?

Excellent

General interest: Is the paper of sufficient general interest?

Good

Quality of the paper: Is the overall quality of the paper suitable?

Excellent

Is the length of the paper justified?

Yes

Should the paper be seen by a specialist statistical reviewer?

No

Do you have any concerns about statistical analyses in this paper? If so, please specify them explicitly in your report.

No

It is a condition of publication that authors make their supporting data, code and materials available - either as supplementary material or hosted in an external repository. Please rate, if applicable, the supporting data on the following criteria.

Is it accessible?

N/A

Is it clear?

N/A

Is it adequate?

N/A

Do you have any ethical concerns with this paper?

No

Comments to the Author

This is an ambitious study and an excellent contribution to the plant speciation literature. I have provided thorough comments in the attached review. Overall, I believe that the theoretical justification for determining the relative genetic and plastic contributions to FT needs to be more thoroughly flushed out. Additionally, while I find your interpretation of the countergradient

pattern intriguing, I believe that effort also need to be devoted to ecological and methodological explanations for the pattern (please see attached Major Comments).

Decision letter (RSPB-2020-3076.R0)

29-Jan-2021

Dear Dr Sianta,

Your manuscript has now been peer reviewed and the reviews have been assessed by an Associate Editor. The reviewers' comments (not including confidential comments to the Editor) and the comments from the Associate Editor are included at the end of this email for your reference. As you will see, the reviewers and the AE are all agreed that this is an impressive and potentially important study, and that the revised version is much improved. However both reviewers have raised some concerns with your manuscript, and we would like to invite you to revise your manuscript to address them.

We do not allow multiple rounds of revision so we urge you to make every effort to fully address all of the comments at this stage (with the exception of Reviewer 2's suggestion to add a comparison to flowering time differences in the field, which I leave to your decision). Also, both reviewers commented that they were unable to access the archived data, so please check this. If deemed necessary by the Associate Editor, your manuscript will be sent back to one or more of the original reviewers for assessment. If the original reviewers are not available we may invite new reviewers. Please note that we cannot guarantee eventual acceptance of your manuscript at this stage.

Research ethics:

Use of animals and field studies:

If your study uses animals please include details in the methods section of any approval and licences given to carry out the study and include full details of how animal welfare standards were ensured. Field studies should be conducted in accordance with local legislation; please

include details of the appropriate permission and licences that you obtained to carry out the field work.

It is a condition of publication that you make available the data and research materials supporting the results in the article (<https://royalsociety.org/journals/authors/author-guidelines/#data>). Datasets should be deposited in an appropriate publicly available repository and details of the associated accession number, link or DOI to the datasets must be included in the Data Accessibility section of the article (<https://royalsociety.org/journals/ethics-policies/data-sharing-mining/>). Reference(s) to datasets should also be included in the reference list of the article with DOIs (where available).

If you wish to submit your data to Dryad (<http://datadryad.org/>) and have not already done so you can submit your data via this link [http://datadryad.org/submit?journalID=RSPB&manu=\(Document not available\)](http://datadryad.org/submit?journalID=RSPB&manu=(Document%20not%20available)), which will take you to your unique entry in the Dryad repository.

Please submit a copy of your revised paper within three weeks. If we do not hear from you within this time your manuscript will be rejected. If you are unable to meet this deadline please let us know as soon as possible, as we may be able to grant a short extension.

Thank you for submitting your manuscript to Proceedings B; we look forward to receiving your revision. If you have any questions, please do not hesitate to get in touch.

Best wishes,
Professor Loeske Kruuk
<mailto:proceedingsb@royalsociety.org>

Associate Editor

Comments to Author:

We now have two reviews of the previous revision. Both reviewers agree and I agree that the paper has improved greatly. However both reviewers have additional comments which I believe will help the authors to clarify many theoretical assumptions and expectations in this work. The

complexities of the system are not trivial and many of the reviewers comments should aid the authors in improving the current manuscript.

Reviewer(s)' Comments to Author:

Referee: 3

Comments to the Author(s).

I agree with the AE and previous reviewers that this manuscript is an important contribution to our understanding of repeated adaption and shifts in flowering time, especially because of the impressive number of pairs included in the study. The manuscript presents some interesting and unexpected results, and it was substantially improved during revision. For example, figure 1 is much clearer now, and uncertainly has been successfully added to the other figures. The authors also improved the manuscript by switching to "genetic/plastic" terminology and by setting up counter gradient selection in the introduction.

That said, I do have a few additional suggestions to further improve the manuscript's clarity (these are detailed below). Also, I think that the authors should more thoroughly respond to reviewer 1's comment about whether the later flowering is representative of field conditions in the main text. For example, they should mention that 11 of 12 previous studies found a shift to earlier flowering in the serpentine habitat. Finally, I'm not an expert on serpentine habitats, but I wonder whether greater competition limiting resource acquisition in the non-serpentine habitat could explain the discrepancy between the previous studies and this one.

Specific comments:

line 72-73: This prediction could be set-up more clearly. For example, that pattern could be caused because taxa with plastic barriers don't progress through the speciation continuum or because of canalization/counter gradient selection. It might be helpful to incorporate this entire paragraph into the following one that unpacks plasticity a little bit more.

line 113: I would point out the pairs that include tolerator species somewhere in the main text.

line 175: Why not include these results (and the results of the analysis that included germination timing) in the supplementary materials?

line 201: Specify that this is the absolute value

line 249: Is this also based on absolute values?

line 288-289: I would include language that points out that this result isn't very statistically robust here. E.g., "sister taxa pairs with more divergent soil environments tended to have more divergent FTs"

line 294: This is also a marginally significant effect.

line 320-322: It would be helpful to see the flowering time distributions for each pair in both soil types in the supplementary material.

Appendix line 81-83: Please include citations.

Referee: 4

Comments to the Author(s).

This is an ambitious study and an excellent contribution to the plant speciation literature. I have provided thorough comments in the attached review. Overall, I believe that the theoretical

justification for determining the relative genetic and plastic contributions to FT needs to be more thoroughly flushed out. Additionally, while I find your interpretation of the countergradient pattern intriguing, I believe that effort also need to be devoted to ecological and methodological explanations for the pattern (please see attached Major Comments).

Author's Response to Decision Letter for (RSPB-2020-3076.R0)

See Appendix B.

Decision letter (RSPB-2020-3076.R1)

17-Mar-2021

Dear Dr Sianta

I am pleased to inform you that your manuscript entitled "Parallel evolution of phenological isolation across the speciation continuum in serpentine-adapted annual wildflowers" has been accepted for publication in Proceedings B.

Data Accessibility section

Open Access

Paper charges

Thank you for your excellent contribution. On behalf of the Editors of the Proceedings B, we look forward to your continued contributions to the Journal.

Yours sincerely,
Professor Loeske Kruuk
Editor, Proceedings B
mailto: proceedingsb@royalsociety.org

Associate Editor:
Board Member

Comments to Author:

Thank you for a very thoughtful and revision and extensive response to the reviewers. I am confident that this work will make an important contribution to the literature on parallel evolution and also the serpentine grassland study system.

Appendix A

Dear AE,

We would like to sincerely thank you and the two anonymous reviewers for careful and insightful comments. We have accepted the suggests of the vast majority of comments and they have substantially improved the flow and clarity of the article. For example, we have simplified our conceptual Figure 1 to provide a more straightforward depiction of our experimental design and analysis. We use more consistent and simplified language throughout the paper for our response variables, model effects and treatments. We modified the introduction to include a broader introduction of the role of plasticity in reproductive isolation, as well as to smooth the transitions among the main conceptual issues we cover. Overall, these revisions make the article more accessible for readers.

We have addressed all general and specific comments in black text below. Where appropriate, we have included the new line numbers for changes, corresponding to the cleaned documents.

Best wishes,
Shelley A. Sianta and Kathleen M. Kay

Comments from the AE:

We have now received two reviews of the paper entitled: "Parallel evolution of phenological isolation across the speciation spectrum in serpentine-adapted plants." Both reviewers and I all agree there is a very important contribution in this paper--seventeen species/population pairs and a very elegant design. The main issue, highlighted very well by reviewer 2 is that the main message of the paper is clear, but the connection between the conclusion and the results if very complicated and difficult for the reader. Figure 1 is a good start but it needs to be massively simplified and clarified such that the reader grasps 1) the hypotheses in the design, and 2) how the data will be used to test the different hypotheses. Reviewer 2 also highlights a number of possible improvements. I agree with all of reviewer 2's extensive and thoughtful comments, especially on the confusing presentation of Figure 1. I would add a few details: uncertainty is not conveyed very well in the figures--Figure 2 and 4 should have some uncertainty measure (e.g. SE) associated with each point. The authors should also check that their taxonomy is up-to-date. I believe the Jepson Herbarium has now moved the *Mimulus* species studied to *Erythranthe*.

- We agree that Reviewer 2 provided a lot of thoughtful and constructive comments, and we have taken their suggestion on nearly all of them. We simplified Figure 1 and think that it now depicts a more straightforward conceptual walk-through of our experimental design and how we used the data. We included a last sentence in the Figure 1 caption that describes our directional hypothesis for how we predict endemic and tolerator pairs will differ in flowering time divergence.
- We incorporated measures of uncertainty in Figures 2 and 4. Figure 2 now depicts the mean and 95% credible intervals from our Bayesian model of mean divergence in flowering time. Figure 4 now includes 95% confidence intervals from bootstrapped estimates of phenological isolation.

- We respectfully choose to stick with the *Mimulus* taxonomy (i.e., *Mimulus nudatus* and *Mimulus guttatus* instead of *Erythranthe nudata* and *Erythranthe guttata*) because of the reasons presented by Lowry et al., 2019. “The case for the continued use of the genus name *Mimulus* for all monkeyflowers.” *Taxon*, 68(4). The *Mimulus* taxonomy is widely recognized and the *Erythranthe* taxonomy remains controversial. However, we are flexible on this matter.

Last, the countergradient result was a surprise when I got to it, since it was not mentioned in the abstract. The related question about whether serpentine soils truly are drier than nearby soils also deserves some attention. I'm not sure there is evidence to this effect. It's possible that serpentine soils are not actually drier, but that they "act" drier in late spring in some respects because of the relative growth rates of root systems on and off of serpentine, with the root systems on ultra-mafic soils slower to develop because of the overall slower rates of growth and carbon capture. Thus, by the time late spring occurs the root systems of non-serpentine individuals are set up to continue water uptake into May, while on the serpentine soils they are not large and extensive enough to continue. Field et al. (1997) and Hull and Wood (1984) might be interesting to look at from this perspective although this is not the main focus of either paper. Moreover, one of the points in the Field paper is that serpentine soils may in fact stay wetter, all else equal, because there is less leaf area to transpire water. All in all, it may be interesting to consider the countergradient results in light of this alternative hypothesis.

Field, Christopher, et al. "CO2 effects on the water budget of grassland microcosm communities." *Global Change Biology* 3.3 (1997): 197-206.

Hull, James C., and Sarah G. Wood. "Water relations of oak species on and adjacent to a Maryland serpentine soil." *American Midland Naturalist* (1984): 224-234.

- We added a sentence about the possibility of countergradient selection shaping plasticity early in the introduction (New Lines 83-85), and a sentence at the end of the introduction about our specific approach to explore whether and how plasticity has evolved following edaphic divergence (New Lines 133-135). We choose to not highlight the countergradient selection hypothesis in the abstract because of space limitations.
- Regarding whether serpentine soils are wetter or drier than nonserpentine soils – the evidence on this is mixed, as you note. For example, in a study across multiple *Streptanthus* (Brassicaceae) taxa, Cacho and Strauss (2014) show that the serpentine soils *Streptanthus* grows on are rockier (and likely dry out more quickly) than the nonserpentine soils *Streptanthus* grows on. A study by Dittmar and Schemske (2017) measured soil moisture content in adjacent serpentine and nonserpentine populations of *Leptosiphon parviflorus* and found that the serpentine soils dry out more quickly than nonserpentine soils.

CACHO, N.I., and S.Y. STRAUSS. 2014. Occupation of bare habitats, an evolutionary precursor to soil specialization in plants. *Proceedings of the National*

Academy of Sciences, USA 111: 15132–15137.

DITTMAR, E.L., and D.W. SCHEMSKE. 2017. The edaphic environment mediates flowering-time differentiation between adjacent populations of *Leptosiphon parviflorus*. *Journal of Heredity* 109: 90–99.

- In a pooled comparison of all the serpentine and nonserpentine soils in our study, we find that the serpentine soils have on average higher percent sand composition (as opposed to clay or silt) than the nonserpentine soils, suggesting that they may also dry out more quickly (Sianta and Kay, 2019).

Sianta SA, Kay KM. 2019 Adaptation and divergence in edaphic specialists and generalists: serpentine soil endemics in the California flora occur in barer serpentine habitats with lower soil calcium levels than serpentine tolerators. *Am. J. Bot.* 109, 1–14.

- The mix of evidence supports the common assumption that serpentine soil involves more water stress than nonserpentine soil – whether due to drier soils *per se* or limited water uptake by the plant. We added text about this in the 4th paragraph of discussion (New Lines: 376-379), where we highlight the result that most serpentine plants flowered later than their nonserpentine sister, contrary to the common assumption.

Comments from Reviewer 1

This interesting paper describes a novel and significant experiment examining the role of genetic differentiation and phenotypic plasticity of flowering phenology in shaping reproductive isolation of sister taxa during colonization of serpentine soils. The strength of the study is the use of 17 serpentine/nonserpentine sister species pairs on native and non-native soils in a large greenhouse experiment. The authors find that genetically based phenological shifts and the degree of phenotypic plasticity both occur, but do not differ substantially between sister species pairs vs. within-species pairs of serpentine/nonserpentine ecotypes (contrary to their initial prediction that plasticity would be more important earlier in the process of taxon divergence). A surprising result from this study is that in the majority of taxa pairs serpentine taxa flower later than their non-serpentine sisters- even under common garden conditions. This finding is contrary to conventional wisdom that serpentine taxa flower earlier in the field and surprising in light of the phenotypic selection for earlier flowering on serpentine soil observed in this experiment.

In general, I am enthusiastic about this paper. The experiment is well designed, the results are interesting (albeit surprising), and it is well-written. However, I do have a few questions and comments.

Lines 101-103: This is conventional wisdom, but in how many species/populations has it actually been shown? Do the authors have field observations of approximate flowering phenology – or at least seed collection dates- for the taxa in this study? If so, do those observations show similar patterns to the results of the greenhouse experiment? Given the surprising finding of delayed flowering on serpentine, it would be good to show that this pattern is representative of field populations.

- I collated a list of 12 papers that described flowering time differences in between serpentine-nonserpentine sister taxa, and 11 of them showed a shift to earlier flowering. Below are the references to this list. We added representative, but not comprehensive, citations to the first clause in this sentence. (New Lines: 99-101)

ARNOLD, B.J., B. LAHNER, J.M. DACOSTA, C.M. WEISMAN, J.D. HOLLISTER, D.E. SALT, K. BOMBLIES, and L. YANT. 2016. Borrowed alleles and convergence in serpentine adaptation. *Proceedings of the National Academy of Sciences* 201600405.

DITTMAR, E.L., and D.W. SCHEMSKE. 2017. The edaphic environment mediates flowering-time differentiation between adjacent populations of *Leptosiphon parviflorus*. *Journal of Heredity* 109: 90–99.

GAILING, O., M.R. MACNAIR, and K. BACHMANN. 2004. QTL mapping for a trade-off between leaf and bud production in a recombinant inbred population of *Microseris douglasii* and *M. bigelovii* (Asteraceae, Lactuceae): A potential preadaptation for the colonization of serpentine soils. *Plant Biology* 6: 440–446.

GARDNER, M., and M. MACNAIR. 2000. Factors affecting the co-existence of the serpentine endemic *Mimulus nudatus* Curran and its presumed progenitor, *Mimulus guttatus* Fischer ex DC. *Biological Journal of the Linnean Society* 69: 443–459.

KRUCKEBERG, A.R. 1950. An experimental inquiry into the nature of endemism on serpentine soils. University of California, Berkeley.

MACNAIR, M., and M. GARDNER. 1998. The evolution of edaphic endemics. In D. Howard, and S. Berlocher [eds.], *Endless forms - species and speciation*, 157–171. Oxford University Press, New York City, New York, USA.

MURREN, C.J., L. DOUGLASS, A. GIBSON, and M.R. DUDASH. 2006. Individual and combined effects of Ca/Mg ratio and water on trait expression in *Mimulus guttatus*. *Ecology* 87: 2591–2602.

O'DELL, R.E., and N. RAJAKARUNA. 2010. Intraspecific variation, adaptation, and evolution. In S. P. Harrison, and N. Rajakaruna [eds.], *Serpentine: The Evolution and Ecology of a Model System*, 97–138. University of California Press, Berkeley, CA, USA.

SAMBATTI, J.B.M., and K.J. RICE. 2007. Functional ecology of ecotypic differentiation in the Californian serpentine sunflower (*Helianthus exilis*). *New Phytologist* 175: 107–119.

SCHMITT, J. 1983a. Density-dependent pollinator foraging, flowering phenology, and temporal pollen dispersal patterns in *Linanthus bicolor*. *Evolution* 37: 1247–1257.

SCHMITT, J. 1983b. Individual flowering phenology, plant size, and reproductive

success in *Linanthus androsaceus*, a California annual. *Oecologia* 59: 135–140.
WRIGHT, J.W., M.L. STANTON, and R. SCHERSON. 2006. Local adaptation to serpentine and non-serpentine soils in *Collinsia sparsiflora*. *Evolutionary Ecology Research* 8: 1–21

- Comparing our greenhouse phenology data with field data is a great idea. We do have preliminary data on field observations (e.g., seed collection dates), although we hope to combine that data with more detailed data of field phenology patterns for a later paper.

Lines 177-178. Is time from germination to flowering the most ecologically relevant metric? Did the seeds all germinate at the same time? Or were there differences within and among sister taxa pairs in days to germination? Were there differences in germination timing between soil types, which might have different hydrothermal properties? If so, then flowering date in Julian days would depend upon both germination and flowering phenology, and the reported metric might not tell the whole story. Hopefully this was not an issue, but the authors need to address this by explicitly discussing variation in germination phenology. It would also be useful to know a little more about the seasonal timeline of the experiment (e.g. watering timing) and how it aligned with the phenology of natural populations. Some of this is in the SI, but please point the reader to it.

Variation in germination timing

- There were no, or subtle, differences in germination timing among treatments within a sister taxa pair. The sister taxa pairs for which there were significant differences in germination timing among treatments had subtle differences – e.g., one treatment that varied from the other treatments by, on average, under 3 days. One endemic pair, the *Collinsia greenei-sparsiflora* pair, had the serpentine taxon germinating later than the nonserpentine taxon (by on average 11 days in the serpentine soil) – however, because the serpentine taxon in this pair also flowers later than the nonserpentine taxon (and has one of the larger flowering shifts of any of the pairs), including the germination timing would only increase the magnitude of this shift.
- We also reran the flowering onset analyses with a measure to days-to-flower that incorporated germination timing – results were qualitatively not different than the results that exclude germination timing, and are not presented in the text.
- There was some variation in germination timing among the sister taxa pairs; however, we don't feel that is important, as all of our among-pair analyses use relative differences between the sister taxa within a pair.
- We explicitly address the no-to-subtle variation in germination timing in the methods section (New Lines: 174-176)
- We acknowledge that Julian days, measured in the field, incorporate both variation in germination timing and variation in time from germination to flowering. However, given that there are subtle differences in germination timing (at least as a function of soil, seed source, watering regime, and controlled temperature/light regimes in our experiment), we think it's reasonable to ignore the effects of germination variation in the field. Future

studies documenting flowering time distributions in these populations will incorporate sources of variation in germination timing that aren't captured in our experiment. Comparing those future studies with this greenhouse data will allow us to disentangle the effects that variation in germination timing versus germination-to-flowering timing have on phenological isolation.

More detailed experiment timeline:

- We added more information on the timeline of the experiment in Appendix 2, which originally described the detailed growing conditions (with some information on the experimental timeline). The new information is put into a section labelled "Timeline" and the original material from Appendix 2 is now under the heading of "Conditions." (New Appendix Lines: 105-115)
- In the main text, where we refer to Appendix 2, we added an additional clause that tells readers that the greenhouse timeline information is in Appendix 2. (New Lines: 169-170)

Line 194. An unfortunate typo!

- Indeed! We ended up primarily using different language – "divergence" – in place of "shifts"

Lines 228-237. I found the term "seed" effect to represent the genetic effect of taxon to be extremely confusing. I recommend finding a different terminology, e.g. "taxon" or "native soil" - if using the latter maybe refer to the "soil" effect as "soil treatment".

- Thank you for the suggestion on improving clarity – we have removed the term "seed" effect. Where necessary, we replace "seed" with "taxon," but we primarily stick with "genetic" vs "plastic" effects to describe the effects in the linear model.
- New Lines: 215-235

Lines 278-283: Was there any evidence of non-linear or stabilizing/disruptive selection? Were there any covariates included? If there wasn't enough power for these tests, that's fine, but please say that. More importantly, why only measure selection on serpentine soil? Isn't the hypothesis that selection for early flowering will be stronger on serpentine than non-serpentine? If so, the test requires comparing selection gradients from both environments. I would also like to see a little more about the model used. Was the analysis done with absolute flower production (as implied by the text) or relative flower number per Lande and Arnold 1983?

Linear and non-linear selection analyses:

- Thank you for pointing out the absolute flower production – we had mistakenly used absolute flower production as the response variable in the linear selection analyses. We reran analyses with relative fitness (relative flower production) and changed the corresponding table (Table S6) and figure (Fig S5). Now the selection gradients are comparable across taxa.
- We did not use any covariates in the selection analyses.

- We did not originally run non-linear selection analyses, in part because some taxa have small sample sizes ($n < 15$). We ran non-linear selection analyses (which included one linear and one quadratic term) on relative fitness for all taxa with $n > 4$ (our sample size cutoff for the linear selection analyses), and we report results in a new Table S2. Three taxa had significant quadratic terms, although only 1 of those was significant after corrections for multiple comparisons. All three taxa had positive quadratic terms, indicative of disruptive selection. However, plotting the quadratic equations over the data shows that the quadratic equations for 2 taxa are largely driven by 1-2 outlier points. The third taxon has a “shallow” quadratic term – i.e., there is little curve to the equation line. Because of the overall little evidence for non-linear selection, we focus on the linear selection analyses in the main text. We refer the reader to the non-linear analyses at the end of the “Method – data analysis” section about phenotypic selection.
 - New line: 270-273

Why only measure selection on serpentine soil?

- We focused on measuring selection only on serpentine soils because we were interested in using selection in that environment to understand whether serpentine-induced plasticity on flowering time was maladaptive.

Line 296- This sentence is not consistent with Figure 2 as I understand it. What happened to the negative shifts shown for some taxon pairs in that figure?

- We were referring to the absolute magnitude in flowering time shifts, which is why there is no directionality in that sentence. To clarify, we changed “Total” to “Absolute”
- New Line: 283

The idea of counter-gradient adaptation in phenology is interesting and plausible. But one interesting point that isn't really mentioned is that there's no evidence for selection for phenological reinforcement of reproductive isolation. That seems worth a mention.

- This is an interesting point, as there's at least one example of possible reinforcing selection on flowering time driven by selection against hybrids at another edaphic boundary (Silvertown et al., 2005. “Reinforcement of reproductive isolation between adjacent populations in the Park Grass Experiment.” *Heredity* 95:198-205.).
- We don't discuss reinforcing selection in the context of counter-gradient variation, because we don't have the appropriate design to detect reinforcing selection due to edaphic divergence– e.g., independent sets of sympatric and allopatric populations for a given sister taxa pair and ecologically-dependent hybrid fitness. We feel that adding text about the possibility is too speculative. These systems of secondary contact along edaphic boundaries are a worthy area to pursue investigations of reinforcing selection.

Comments from Reviewer 2:

In this paper Siantis and Kay test whether repeated adaptation to serpentine soil both within and between species is associated with repeated genetic and plastic differences in phenology measured by flowering time (onset). The authors use species from across a broad taxonomic scope which makes the paper quite interesting. The authors used rigorous and appropriate statistical analyses in which they control for phylogeny to test whether the degree of flowering divergence is associated with evolutionary divergence and reproductive isolation. The authors find that divergence in phenology is a common occurrence when taxa are adapting to serpentine soil and that that divergence is primarily genetic rather than plastic. They find limited evidence for genetic compensation in flowering time.

I feel that while there are interesting results in this paper, it could be significantly improved in several ways. First of all the introduction and other parts of the text are somewhat confusingly written. In the introduction there is a lot of switching back and forth between talking about the roles of plastic vs genetic phenological isolation which makes it difficult to follow the exact big picture question that the authors are trying to answer. In general a lot of sections are wordy which makes them more difficult to follow and confusing, wordy names are used to identify treatments and effects. For example why use seed and soil effects when you mean genetic and plastic effects? See the specific comments below for more examples. Simplifying and clarifying the language used throughout the paper would make it more readable.

- Generally, we improved the clarity of the introduction by introducing the role of plasticity vs genetic differentiation in reproductive isolation as a broader conceptual point. We simplified the language around our treatments and statistical effects throughout the paper. Your specific comments were very helpful and constructive. We address each below.

An experimental design issue that is not addressed in the manuscript is using maternal families rather than inbred lines to measure plasticity. This introduces a confounding effect since siblings are genetically different. This confounds genetic differences with environmentally induced plastic differences. The authors need to acknowledge this in the paper and justify their experimental design or interpret their results in light of this confounding factor.

- We acknowledge that the difference in trait value between two siblings in different environments reflects both the effect of plasticity and the genetic differences between siblings. Genetic differences between siblings might contribute to variance in flowering time within a soil, but shouldn't bias our estimate of plasticity if there is a consistent, and substantial, plastic effect of soil. That multiple sibling pairs all respond in the same direction in some taxa suggests that the differences between treatments are primarily driven by plasticity. When there is no true, or little, effect of plasticity, it could be that the genetic differences among siblings mask a subtle effect of plasticity. However, in those cases, plasticity is less likely to be biologically relevant.
- We incorporate your suggestions below, in the specific comments, of where to acknowledge the potential effects of sibling genetic variation on our plasticity measures.

Another possible experimental design issue not addressed is life-history variation. Since flowering time is often varies with other life-history traits it would be good to know whether the taxa used here differ in annual perennial life-history? Or are they all annuals? If they do differ (for example if most serpentine endemics are perennials), then this could have a confounding effect on flowering time differences in addition to serpentine soil and therefore it should be controlled for in the models.

- Thank you for pointing this out. All the taxa are annuals, and that information was mistakenly cut out of the main text and placed in the Appendix. We clarified the life history of our taxa both by incorporating “annual wildflowers” into our title (New lines: 1-2), and in the first sentence of the methods. (New line: 139-140).

In general the Figures are confusing. For example Figure 1 is much too text heavy. It makes it look busy and makes it hard to read. The authors should simplify each diagram and remove as much text as possible from the figure. They should use colors and shapes to indicate their meaning instead. For example in Figure 1D there is no need to have the soil icons next to each curve, you’ve already designated the curves as serp or non-serp based on circles and triangles. The icons add busyness. Figure 1C is so complex and confusing I really can’t say what results it is communicating. The authors should try to dramatically simplify at least the first two figures. Specific suggestions are below.

- We used all of your specific suggestions, and more, to simplify and clarify the figures – particularly figure 1. Detailed descriptions of the changes we made are under the corresponding specific comments.

Specific Comments:

Line 83: I’m getting a little confused by the introduction of plasticity here. Why are you switching back and forth between non-plastic and plastic changes in FT? If testing whether plasticity is important for RI is one of the goals of your study that idea should we introduced earlier and in a broader context.

- We added a paragraph in the beginning of the introduction that introduces the broader conceptual ideas of genetically-based versus plastic trait changes in promoting reproductive isolation and speciation (New lines: 67-73).

Line 85-86: This transition right here is what is confusing about switching back and forth between plasticity and non-plastic FT divergence.

- We merged the two phenological isolation paragraphs, and are more explicit about plasticity in phenological isolation near the start of the paragraph. With the addition of

the broader conceptual paragraph of genetically-based vs. plastic contributions to reproductive isolation, we think that the contrasting of the two drivers of trait divergence (i.e., genetic vs plastic change) will be more understandable to the readers. (New lines: 74-90)

Lines 86-88: And where does plasticity fit into this goal? Please clarify.

- We clarified this sentence to show that one of our goals is to document the relative contribution of genetically-based versus plastic trait changes to phenological isolation. (New line: 87-90)

Line 90: do you mean parallel ecological divergence? Similar is a little vague.

- Yes. We changed “similar” to “parallel”. (New line: 92)

Lines 99-101: “Using natural variation in...” I find this phrasing confusing. Please rephrase for clarity.

- In the process of streamlining the introduction, we deleted this sentence.

Lines 108-109: “17 independent replicates of edaphic divergence associated with either ecotypic differentiation” This sentence is wordy and therefore confusing. I think the reader will assume that the edaphic divergence events you’re studying are associated with phenotypic divergence. Otherwise why would you be talking about these taxa? So there doesn’t seem to be a need to say that explicitly here.

- We rephrased the sentence to “In this study, we examine flowering time divergence across 17 independent replicates of edaphic divergence associated with either the evolution of a serpentine tolerator population or a serpentine endemic species.”

Lines 122: I can see that “flowering onset” is a more precise description of flowering time, but as FT is the accepted term in the literature for the trait you are describing I would use that instead. Although the choice of language is ultimately up to the authors, it might make the paper easier to read.

- We changed the usage of flowering onset to flowering time throughout the paper. To clarify to readers that we are talking about flowering onset when we use flowering time, (FT) we introduce flowering onset and add the clause “hereafter referred to as FT” in the second to last paragraph of the introduction. (New lines: 115)

Lines 123-124: What do you mean by habitat features? Be specific.

- We clarified that we specifically mean divergence in edaphic and/or climate factors.
- (New line: 121)

Lines 122-137: It might be helpful to start this paragraph with the big picture question you are addressing, and then move into all the specific subquestions. Otherwise the reader is likely to get a little lost in the specifics. Remind us what you're most excited about!

- Agreed. We start this paragraph off with: "Our main goals are to understand whether parallel ecological divergence results in parallel phenological isolation, and whether progress towards speciation is influenced by the degree to which flowering time divergence is genetically-based versus plastic."
- (New lines: 117-119)

Line 134-135: Assuming this seems a bit presumptuous. Can you justify this assumption further?

- Taking on this assumption assumes that 1) serpentine taxa are derived from nonserpentine ancestors, and 2) that plasticity in a nonserpentine taxon has not evolved since the split-off of the serpentine taxon. While phylogenetic evidence suggests that the former assumption is common (we incorporate this evidence into the sentence), we acknowledge that the second assumption may not be probable. We note in the sentence in the main text that using the nonserpentine sister taxon's plasticity is a *proxy* for ancestral levels of plasticity.
 - (New lines: 130-133)
- We also note that a similar experimental design – including using the traits of an extant sister taxon as a proxy for traits of a common ancestor - has been used in the context of testing for ancestral plasticity in diet-morphs of spadefoot toads (Levis, Isdaner, and Pfennig. 2018. "Morphological novelty emerges from pre-existing phenotypic plasticity." *Nature Ecology & Evolution*, 2: 1289-1297.)

Line 134-137: I am confused as to how you are going to do phenotypic selection specifically on plasticity in only one soil type...I of course read the rest of the paper and now do understand. But I have left this comment here so that you can see how readers might react to your description of your methods upon first reading this section. The description could be simplified/clarified.

- Thank you for pointing out the ambiguity in this sentence. We changed (bolded component) the sentence to read: "We use phenotypic selection analyses in serpentine soil **to ask whether plastic shifts in serpentine soil are adaptive or maladaptive, and to assess whether** plasticity in serpentine taxa may have evolved following colonization of serpentine" (New lines: 133-135)

Line 172-173: Could this have affected your measurements? How controlled are the greenhouses? Were there big environmental differences between the years?

- Some of the greenhouse conditions did vary between years. The details of the growing conditions for the two experimental rounds, and the differences between them, are detailed in Appendix 2, as described in the main text (New lines: 169-170). We grew a mix of tolerator and endemics pairs in each year, so the differences in growing conditions shouldn't drive any differences we see between the endemic and tolerator pairs. Regardless, we use the year-grown-in-greenhouse as a covariate in our analyses to account for these differences in growing conditions.

Line 181: Typo. "We did not hand-pollinated.."

- Fixed to "hand-pollinate"
- (New lines: 178)

Line 191: It seems like a word like "difference" or divergence might be clearer here than shift.

- We agree and use "divergence" instead of shift throughout – e.g., "flowering time divergence" instead of "shifts in flowering time".

Line 194: Typo. Hehe.

- Fixed! 😊 And then subsequently switched to "flowering time divergence" – another benefit of using divergence instead of "shifts"!

Line 197-199: Can you explain more about what these models control for? I'm assuming they ask how much divergence is due to environment after controlling for phylogenetic divergence, but I've never used them so would be useful to say something briefly about how they work here.

- We added a clause to the sentence that says how PGLS uses "the phylogenetic variance-covariance matrix to structure error terms". (New lines: 194)

Line 225-227: "...reflect plasticity." Or they reflect within species/maternal family genetic variation. Or combination of within species genetic and env var. Did you plant genotypic replicates such as inbred lines in each soil type or just species/maternal family replicates? Without true genetic replicates there is a genetic variation between treatments since even siblings are quite genetically different. You need to acknowledge this confounding factor of genetic variation as well as environmental variation in the paper.

- No, we unfortunately weren't able to create inbred lines for all taxa used in the study. Instead we planted a sibling per maternal family in each of the two soil treatments.
- We acknowledge that using maternal family replication contributes the confounding factor of sibling genetic differences to our plasticity measures at end of first paragraph of Methods data analysis section: "Is flowering time divergence more genetically-based in endemic pairs than tolerator pairs?" (New lines: 222-225)

Lines 231-233: This sentence is very confusing. It would be helpful to come up with simpler, less confusing names for these treatments than "serpentine seed serpentine" etc.

- We rewrote and simplified the sentence in question, as well as another sentence in the same paragraph that used the same long/confusing names for the treatment contrasts. (New lines: 228-231)

Lines 252-254: So does this mean that non-serp taxa were not grown in serp soil? That seems like a missed opportunity. Also only a partial reciprocal transplant...so using the term common garden seems more accurate.

- We indeed grew nonserpentine taxa in the serpentine soil, which is why we use the reciprocal transplant language.
- We did not include reproductive isolation within serpentine soils because the majority of nonserpentine taxa died before flowering in those soils. We are preparing a subsequent manuscript that reports fitness differences among all taxa pairs in the experiment, and will be highlighting the fitness of nonserpentine seeds in serpentine soil.

Lines 257-259: Ah yes that makes sense. However still not quite a "reciprocal" transplant then.

- As mentioned above, because we did transplant the nonserpentine taxon into serpentine soils, we still use the reciprocal transplant language.

Line 267-268: So this isn't a purely "plastic" response since siblings have genetic differences...

- Correct, we insert the following sentence into this section of the methods: "We note that maternal family reaction norms include genetic differences between the two siblings, and with this caveat, use the term "plasticity" to refer to the maternal family reaction norms." (New lines: 257-259)

Lines 297-298: Interesting. Later is not what I would have expected.

Line 299: And soil distance is characterized by the ionic composition of the soil? By the particle size? By temperature and water measurements in the native habitats?

- In the methods section, (New lines: 191-193) we briefly say that we quantify divergence in “soil chemistry and texture.” The full suite of soil chemistry and texture variables that were quantified are detailed in Appendix 3 (New lines: Appendix 149-156), as cited in the methods on New lines: 191-193

Lines 299-303: How does physical distance affect your flowering results? Was it controlled for in these analyses? There could be isolation by distance where pairs that are further apart from each other have undergone great neutral divergence in FT.

- Given that flowering time is such an important life history stage, we assume that it is under selection in most populations (as long as populations are large enough for selection to be efficacious), and thus don't expect neutral divergence in flowering time to affect patterns of divergence.
- Physical distance among populations may still affect our flowering time results, given that environmental factors other than soil vary among populations. We used multivariate climatic divergence between sister taxa as a way to test the effects of physical distance on flowering time. Climatic divergence explained little variation in flowering time divergence (Figure S2), although we had a skewed distribution of climatic [and physical] distance – the majority of our sister taxa are physically proximate to one another (seen in Table S1).

Line 316-319: WHY not just call these genetic and plastic effects? The seed and soil nomenclature is confusing and unnecessary.

- We changed the nomenclature to genetic and plastic effects instead of seed and soil effects, respectively, throughout the manuscript - in both the methods (New lines: 215-235) and results (New lines: 300-307) sections.

Lines 330-333: Doesn't this imply that endemic pairs have a greater genetic divergence in flowering time than tolerator pairs? That seems to contradict what you found above in respect to plasticity. I'm confused how the two analyses are different...

- The results in this paragraph describes permanence of phenological isolation, which incorporates both flowering onset and flowering duration – it's the comparison of the full flowering time distributions of the two sister taxa. We found that endemics have slightly more permanent (less plastic) phenological isolation. That our phenological isolation

results differ from the flowering onset results suggests there is some soil-induced plasticity in flowering duration. We made it more obvious that we measured reproductive isolation using full flowering distributions, and we hope this will clarify the confusion here.

Line 340: You need to acknowledge in the results that there may also be genetic differences between siblings in addition to plastic differences wrought by the environment.

- In addition to the acknowledgment there that the maternal family reaction norms are a proxy for plasticity in the methods (New lines: 257-259), we also add a clause to the first sentence in this paragraph to remind readers that maternal family reaction norms are a proxy for plasticity (New lines: 326).

Lines 346-349: This is an interesting trend though!

Lines 351-355: So the 7 out of 17 was before multiple testing correction? That is not quite clear from how you've presented thing here.

- That is correct – the 7 out of 17 was before multiple testing correction. We changed the language in the sentence to make it more clear that the 7/17 was before the multiple testing correction. (New lines: 338-340)

Lines 362-363: I feel that because your experiment was not truly “reciprocal” between each soil type you should call it a common garden, rather than a reciprocal transplant. That seems like the most accurate term.

- We disagree, because the design was indeed reciprocal – for each taxa pair, the serpentine and nonserpentine taxa were each planted into their native soil and the foreign/opposite soil. We can see how this is misleading given that we do not report phenological isolation in serpentine soils. However, we do use the flowering time data of nonserpentine taxa in serpentine soils to calculate maternal family-level reaction norms.

Lines 366-367: Did you measure reproductive isolation due to flowering time? I don't remember reading about it in the results...

- Yes, we did, and your confusion highlights a lack of clarity on our part. The measurements of phenological isolation (Methods (New lines: 237-251), and Results (New lines: 309-322)) are traditional measurements of this form of reproductive isolation.
- We added clarity to the phenological isolation measure by describing how it is different than the flowering time divergence analyses in the Methods data analysis section for phenological isolation (New lines: 238-240).

Lines 392-394: Yes that makes sense. Does it have anything to do with variation in life-history

of the taxa? Were there perennials in your dataset? And if so do you control for the effect of life-history

- As mentioned above, all taxa we used in this experiment were annuals.

Line 396-399: I am not sure I agree. Just because a closely related extant taxon has a characteristic does not mean that the most recent common ancestor of sister taxa shared that characteristic. It would be more conservative to say it is possible or probable that the plasticity is ancestral, but to assume it seems like a strong and not very scientific statement. I would rephrase.

- We rephrased the sentence to say that “it is possible that the serpentine-induced plasticity in the sister nonserpentine taxon represents an ancestral-like condition in each pair” (New lines: 383-385). We also cite the Levis et al 2018 paper as an example where a similar assumption is made (New lines: 385).
 - Levis, Isdaner, and Pfennig. 2018. “Morphological novelty emerges from pre-existing phenotypic plasticity.” *Nature Ecology & Evolution*, 2: 1289-1297.

Lines 413-415: Was this RI measurement in the results? I don't see it...maybe it was called something else?

- Yes, it is the 2nd to last section in the results, (New lines: 309-322).

Lines 424-426: Are there plans to do a follow up study in the field? And with inbred lines? That would be ideal!

- Not yet ☺ But that would be ideal to nail down the plastic effect, induced by the multivariate field environment.

Table 1: The genus's that have both a within and between species comparison seem particularly worthy of answering these questions. It might be worth highlighting their results separately if they're interesting.

- We agree that the pairs we have nested within genera are particularly useful to answer these questions, and there are some interesting preliminary patterns within the genera that are worth following up on. However, we are short on space in the main text and thus don't elaborate there.

Figure 1: In general this is a lot of text in this figure making it busy and complex looking. Find a way to show things visually with colors/shapes whenever possible rather than adding additional

text and labels.

Figure 1 PartB: I would call this a common garden. Also it might be easier to visualize the setup in B if you had a space divided into quadrants like a punnet square instead of the current diagram. This diagram is a bit visually complex

Figure 1 Part C: Figure 1C is so complex and confusing I have no idea what data it is communicating. Please dramatically simplify.

Figure 1 Part D: Too busy!!! is no need to have the soil icons next to each curve, you've already designated the curves as serp or non-serp based on circles and triangles.

- Figure 1 comments: Thank you for the suggestions with improving clarity. We took all of your suggestions, and simplified panel 1C. We also added more text to the figure legend to help guide readers through the figure.

Figure 2: Again you don't need so much text. Indicate in a simpler Figure legend or in the caption that color indicates tolerator vs. endemic and the shape indicates soil type.

- We simplified this figure by removing the text that was on the main part of the figure, simplifying the legend, and including more detail and clarity in the figure description.

Figure 3: Nice. This is an easy to read figure.

Appendix B

Dear AE,

We would like to thank the two anonymous reviewers for careful and insightful comments. We have accepted the suggestions of the vast majority of comments and they have improved the clarity of the article. We have better motivated our prediction of expecting more genetically-based reproductive isolation in taxa further along the speciation continuum, and clarify the rationale for how we parse out the genetic vs. plastic contributions to total flowering time shifts. We addressed the limitations of a greenhouse experiment, particularly the role that disrupting the soil structure may have on influencing patterns we see in the field vs. the greenhouse. We compiled field observations and seed collection dates for all of our taxa in a new supplementary appendix and, while they only give us resolution on a coarse-scale, they do confirm that the greenhouse patterns we see mimic field patterns in taxa pairs that have marked shifts in flowering time. We incorporated additional analyses and figures in the supplement to confirm that germination variation does not affect our results. Lastly, we clarified the purpose of our discussion that contains the concept of countergradient selection. We feel that these major revisions, as well as myriad minor revisions, have greatly improved the clarity and intention of this manuscript and that readers will enjoy the comprehensive story we tell.

We have addressed all general and specific comments in black text below. Where appropriate, we have included the new line numbers for changes, corresponding to the cleaned (not tracked) documents.

Best wishes,
Shelley A. Sianta and Kathleen M. Kay

AE Comments:

We now have two reviews of the previous revision. Both reviewers agree and I agree that the paper has improved greatly. However both reviewers have additional comments which I believe will help the authors to clarify many theoretical assumptions and expectations in this work. The complexities of the system are not trivial and many of the reviewers comments should aid the authors in improving the current manuscript.

Referee: 3

Comments to the Author(s).

I agree with the AE and previous reviewers that this manuscript is an important contribution to our understanding of repeated adaptation and shifts in flowering time, especially because of the impressive number of pairs included in the study. The manuscript presents some interesting and unexpected results, and it was substantially improved during revision. For example, figure 1 is much clearer now, and uncertainly has been successfully added to the other figures. The authors also improved the manuscript by switching to "genetic/plastic" terminology and by setting up counter gradient selection in the introduction.

That said, I do have a few additional suggestions to further improve the manuscript's clarity (these are detailed below). Also, I think that the authors should more thoroughly respond to reviewer 1's comment about whether the later flowering is representative of field conditions in the main text. For example, they should mention that 11 of 12 previous studies found a shift to earlier flowering in the serpentine habitat. Finally, I'm not an expert on serpentine habitats, but I wonder whether greater competition limiting resource acquisition in the non-serpentine habitat could explain the discrepancy between the previous studies and this one.

We are already at the word limit, which includes citations, and there is simply no way to include 12 new citations without drastically cutting the paper. Moreover, there are many reviewer comments about adding text, but no suggestions for text to cut. We feel that the issue of field flowering time differences is too complex to be treated fairly within this paper, since there are potentially confounding effects of latitude and elevation, as well as non-trivial issues of data filtering from herbarium specimens and other public databases. In our prior draft, we cited two qualitative reviews that cover most of these studies. We now cite some of the prior studies as examples, and change the sentence, “Moreover, shifts in FT are commonly noted in serpentine systems [33,39], with shifts to earlier flowering in annuals hypothesized as a way to escape the drought-inducing conditions of rocky serpentine soils, although it is not clear to what extent reported patterns are due to water holding capacity or serpentine chemistry *per se* as these are often related [34,40,41]” to “Moreover, shifts in FT are commonly noted in annual serpentine systems [31,33,34,39,40], with earlier flowering the most common pattern reported. Earlier flowering is hypothesized as a way to escape the drought-inducing conditions of rocky serpentine soils. Alternatively, life history theory predicts that plants in stressful habitats should flower later because of resource constraints.” This change also better sets up our results, which we strongly believe are not a greenhouse artifact based on our field notes and seed collection dates.

We note in the Discussion (Lines 436) that the lack of competitive environment may alter plastic effects on flowering time that we see in the greenhouse vs. the field. Field observations (Appendix 8) and measures of bare ground in these populations (Sianta and Kay 2019) actually suggest that occurring in a competitive environment (e.g., the blue oak woodlands, nonserpentine habitats), results in relatively early flowering, likely due to selection to reproduce before the exotic annual grasses take over.

Specific comments:

line 72-73: This prediction could be set-up more clearly. For example, that pattern could be caused because taxa with plastic barriers don't progress through the speciation continuum or because of canalization/counter gradient selection. It might be helpful to incorporate this entire paragraph into the following one that unpacks plasticity a little bit more.

Thank you for pointing this out. All of the processes you mention can lead to the prediction that reproductive isolation will be more strongly genetically-based for taxa further along the speciation continuum.

To make the prediction more clear, we moved up some of the conceptual topics from the 3rd paragraph to this 2nd paragraph. Specifically, we expanded the idea that the ephemeral nature of plastic reproductive isolation (due to environmental change and/or dispersal) can make reproductive isolation break down under different environmental contexts and prevent movement along the speciation continuum. We note that taxa that have had more time to diverge also may have had more time to accumulate genetically-based reproductive barriers. We also moved up the ideas of selection reducing plasticity over time, either through countergradient selection or through canalization (depending on whether plasticity is maladaptive or not, respectively). The addition of these various processes should make it more clear why we expect taxa pairs that are further down the speciation continuum to have more genetically-based barriers. Lines 67-80

line 113: I would point out the pairs that include tolerator species somewhere in the main text.

In Table 1, we added whether the nonserpentine taxon of the endemic pairs comes from a species that is a serpentine tolerator or a serpentine non-tolerator.

line 175: Why not include these results (and the results of the analysis that included germination timing) in the supplementary materials?

We created a new appendix in the supplementary material (Appendix 3) that documents 1) variation in germination within sister taxa pairs and whether there are significant effects of taxon or soil on germination timing, 2) the primary analyses of flowering time divergence including germination variation. We find there is little to no effect of variation in germination timing on our results – flowering time divergence within pairs only varies within 2 of 17 pairs, and variation in germination does not affect the magnitude of flowering time divergence between endemic and tolerator pairs.

Because there was no qualitative effect of incorporating germination timing variation on our main flowering time analyses, we did not redo analyses of 1) flowering time divergence vs climatic or soil multivariate distances, 2) genetic vs. plastic variance components in flowering time divergence, nor 3) differences in maternal family reaction norms between S and NS taxa.

line 201: Specify that this is the absolute value

Thank you for picking this up – we specified that we modelled absolute values.

line 249: Is this also based on absolute values?

Yes, but not in the same sense that we use absolute values for flowering time divergence. For the flowering time divergence analyses, we explicitly calculate absolute value of differences in days-to-flowering-onset between the sister taxa. In contrast, the phenological isolation metrics implicitly incorporate absolute isolation – i.e., the phenological isolation metric would be the same if you were to swap the taxon identities of the two flowering time distributions that are used to calculate phenological isolation.

line 288-289: I would include language that points out that this result isn't very statistically robust here. E.g., "sister taxa pairs with more divergent soil environments tended to have more divergent FTs"

We changed the sentence as suggested. Line 297-298

line 294: This is also a marginally significant effect.

We noted that the effect of pair type is marginally significant. Line 303-304

line 320-322: It would be helpful to see the flowering time distributions for each pair in both soil types in the supplementary material.

We added these graphs to the supplementary material (Figures S4-S6) and referenced those figures in the main text results section (Line 327) and in the methods on Lines 256-257

Appendix line 81-83: Please include citations.

We added the program citations and version numbers for both Mesquite and Muscle. Lines 97-98

Referee: 4

Comments to the Author(s).

This is an ambitious study and an excellent contribution to the plant speciation literature. I have provided thorough comments in the attached review. Overall, I believe that the theoretical justification for determining the relative genetic and plastic contributions to FT needs to be more thoroughly flushed out. Additionally, while I find your interpretation of the countergradient pattern intriguing, I believe that effort also need to be devoted to ecological and methodological explanations for the pattern (please see attached Major Comments).

Major comments:

The authors conduct an ambitious set of greenhouse experiments using 17 different plant taxa pairs that vary in their divergence times to determine how phenological reproductive isolation associated with adaptation to serpentine substrates might contribute to the process of speciation. The authors find in the greenhouse at least, that lineages occupying serpentine substrates consistently flower later than closely related lineages occupying non-serpentine substrates. This finding is counter to the commonly held expectation that plants in California occupying drier habitats (i.e., serpentine) should flower earlier to avoid the impending summer drought. The authors find flowering time shifts, and subsequent phenological isolation, to be an important

barrier early along the speciation continuum, particularly for lineages colonizing inhospitable habitats such as serpentine that are expected to exert selection pressures on life history transitions. They argue that genetically based shifts in flowering time are pervasive among the taxa pairs assessed, and that there is no difference in plasticity among less-diverged population pairs compared to more-diverged species pairs. The finding that phenological isolation is a strong barrier to gene flow among diverging lineages further strengthens the link between abiotic adaptation and the evolution of reproductive isolation in flowering plants. One of the major strengths of this work is its broad taxonomic scope (17 pairs), which allows for a throughout assessment of how the occupation of novel serpentine habitats might result in consistent parallel evolution. This work provides a notable contribution to the plant speciation literature. Overall, the authors have made a great effort to better explain their complex design and data analysis in the re-submitted manuscript; however, I believe that they still need to clarify many theoretical assumptions and expectations throughout the manuscript.

Specifically:

1. The hypothesis that true “species” should have stronger, and/or more genetically based phenological isolation compared to divergent populations is intuitively reasonable; however, this may result from several processes, and carries several assumptions. I believe the authors need to be explicit in their rationale for this hypothesis (see below).

In the second paragraph of the introduction, we better motivate the hypothesis that taxa further along the speciation continuum should have more genetically-based reproductive barriers. As mentioned above (see Reviewer #3’s comment), this hypothesis could be driven by multiple processes, and we highlight multiple of them. We do not single out any of these processes *a priori* as being more important than the other.

2. Lines 124 (Fig. 1C); how the authors are determining the plastic vs. genetic contribution of flowering time divergence is not fully explained or justified here in the text nor in Figure 1. Since the relative contribution of these components is one of the central questions of the study, this section needs to be thoroughly flushed out. Why is the plastic contribution determined via the serpentine taxon in serpentine soil vs. serpentine taxon in nonserpentine soil, and the genetic component as the serpentine taxon in nonserpentine soil vs. nonserpentine taxa in nonserpentine soil? Is this simply logistically convenient (i.e., serpentine species will survive in non-serpentine soils, and not vice versa), or does this have to do with the fairly well-supported hypothesis that non-serpentine is the ancestral state for many serpentine endemics in CA?

Our method for parsing out the genetic vs. plastic contributions to flowering time divergence are primarily driven by the hypothesis that nonserpentine is the ancestral state for many serpentine lineages in CA (as we cite in the text, lines 136-137). Under this hypothesis, flowering time divergence in the field between serpentine and nonserpentine plants (i.e., when they are growing in their own habitats) is driven by the serpentine population diverging in flowering time from a nonserpentine population. We argue that said divergence can be due to genetic change in the serpentine taxon or serpentine-mediated plastic shifts in flowering in the serpentine taxon. To measure the genetic change, we phenotype both taxa in the same ancestral-like nonserpentine

habitat. To measure the plastic change that serpentine soil causes on the serpentine plants, we measure the difference in the serpentine taxon across soil types.

We add text to the beginning of the methods section for this analysis (Lines 223-231) to clarify our rationale for determining the plastic vs. genetic contributions to flowering time divergence.

3. Determination of phenological RI entails understanding the full flowering windows of each species. The authors indicate that depending upon mating system, that their greenhouse treatment (no pollination) may have alternated the natural flowering time distributions of some species. It is unclear how this was handled when calculating overall RI-phenology for each taxa pair.

We did not do anything different in calculating phenological RI as a function of mating system. In all cases, we use the number of open flowers per census day to build flowering time distributions and calculate RI. Because plants that do not get pollinated keep producing flowers for longer durations, we expect that taxa pairs that do not self-pollinate will have more stretched out flowering distributions. Elongation of the distributions, while also comprising more flowers than if the plants were pollinated, leads to more overlap in the distributions and less phenological RI. This underestimation of phenological RI should not bias results for differences in RI between endemics and tolerators, as both types of taxa pairs have both mating systems.

We comment on this in the main text in both the methods (Lines 185-190) and discussion (Lines 437-440)

4. One of the most important findings of the study is that phenological isolation of population pairs is essentially equal to that of “full” species pairs. This suggests that phenological isolation evolves relatively quickly during speciation (at least for lineages in which habitat divergence is associated with speciation). This result rests on the fact that the authors used levels of ITS sequence divergence as proxies for divergence time. This has two limitations, that I believe do not detract from the overall findings of the study, but do warrant discussion. First, how might unequal rates of molecular evolution across different lineages bias their estimates of time since divergences (i.e., is it fair to conclude that, for example, *Clarkia* and *Mimulus* taxa pairs with similar rates of ITS sequence divergence represent similar places along the speciation continuum?). I commend the authors for mostly calculating ITS sequence divergence from the very same accessions used in their experiments. The species pairs and population pairs used in this study seem to show substantially different levels of ITS divergence, suggesting that the “species” and “population” delimitations in the study are not the result of taxonomic inconsistencies between groups. Much of this difference however seems to be driven by the extremely high sequence divergence observed in a single species pair (COGR – COSP). I believe the authors should at least verbally address the ramifications of this, as one of their major findings seem to be that phenological RI among “young” lineages (low ITS divergence) is essentially the same as that of “older” lineages (high ITS divergence), thus is at least partially casual for speciation.

The reviewer is correct that ITS sequence divergence is a somewhat crude estimate of divergence time. However, without a good fossil record and/or detailed population genetic analyses of all pairs, it is the best estimate of divergence time possible, and we stay away from using divergence time among pairs as a quantitative predictor of any response variables in this paper. Previous studies have shown that rates of evolution of ITS vary with herbaceous v. woody or annual v. perennial lifeforms (see citations below), but we restricted this study to only annuals. Also, although one species pair has an unusually high amount of divergence, if we remove that pair, endemic species pairs still have significantly higher sequence divergence than tolerator population pairs (t-test: $t = -2.2$, $df = 14$, $p = 0.043$). Moreover, the species with overlap between the endemic and tolerator categories (in which one species of the endemic pair comprises a tolerator pair) show sequence divergence patterns qualitatively consistent with their taxonomic status. Finally, there is no reason to think that the taxonomic splitting/lumping of these lineages varies in a systematic way with phenological divergence, as species are all circumscribed based on multiple morphological characters. We feel confident that our pairs represent real differences between levels of taxonomic divergence based on the long history of careful taxonomy of these clades. Because of space limitations, we don't feel that this warrants more discussion in the main manuscript.

Citations:

- Richardson, James E, R Toby Pennington, Terence D Pennington, and Peter M Hollingsworth. "Rapid Diversification of a Species-Rich Genus of Neotropical Rain Forest Trees." *Science* 293, no. 5538 (September 2001): 2242–45.
- Malcomber, S T. "Phylogeny Of Gaertnera Lam. (Rubiaceae) Based on Multiple DNA Markers: Evidence of a Rapid Radiation in a Widespread, Morphologically Diverse Genus." *Evolution* 56, no. 1 (January 2002): 42–57.
- Kay, K M, Justen B Whittall, and Scott A Hodges. "A Survey of nrDNA ITS Substitution Rates across Angiosperms Reveals an Approximate Molecular Clock with Life History Effects." *Submitted To Molecular Phylogenetics and Evolution*, n.d.
- Andreasen, K, and BG Baldwin. "Unequal Evolutionary Rates between Annual and Perennial Lineages of Checker Mallows (Sidalcea, Malvaceae): Evidence from 18S-26S rDNA Internal and External Transcribed Spacers." *Molecular Biology and Evolution* 18, no. 6 (2001): 936–44.

5. The authors find a pattern in which serpentine lineages, when grown in the greenhouse, begin to flower much later than closely related lineages from more hospitable nonserpentine sites (presumably wetter soils). The authors' discussion of this results revolves almost entirely around the idea of countergradient variation. I believe a broadened discussion here would add to the manuscript. Specifically, how might serpentine soils in the greenhouse exert different selection pressures than serpentine sites in the field (site shade/sun; field water holding capacity given presence of rocks vs. greenhouse water holding capacity in the absence of rocks). The authors have previously quantified soil texture in at least some of the same taxa pair, and may be able to discuss these data further in the context of their greenhouse findings. I'm curious to know whether the countergradient pattern is methodological (related to soil structure)? Alternatively, and I am somewhat hesitant to suggest this as the authors have clearly

devoted a considerable amount of effort to this study already, but I would love to see a comparison to flowering time differences in the field. One potentially feasible idea here would be to download georeferenced herbarium records for all species in the analysis, and subset records based on their occurrence on or off serpentine soils. Next, while not without some limitations, the authors could then extract collection dates for each record, and construction collection date (i.e., flowering time) distributions for each species pair and then calculate phenological RI based on these curves. This would allow the authors to assess whether RI resulting from flowering time shifts in the field were consistent with their findings from the greenhouse, or whether countergradient variation under experimental conditions obscures or weakens this reproductive barrier in nature.

The potential differences between field and greenhouse conditions that could affect flowering time certainly warrant discussion. Ideally, we want to know if the order of flowering time (do serpentine or nonserpentine plants flower first?) is the same in the field as in the greenhouse, given that our greenhouse data contrast with patterns from the serpentine literature.

While a robust documentation of flowering time differences in the field would be an ideal comparison for our greenhouse data, we do not quantify field flowering time differences with herbaria records. First, there are several limitations to using herbaria records for documenting fine scale flowering time distributions: records may not have been collected during flowering and/or there is often no documentation of what stage of flowering the population is at, allowing for differentiation of flowering time distributions only at relatively coarse time scales (e.g., weeks-months). Given that the majority of our pairs experience moderate phenological isolation (with substantial overlap), we would not expect herbaria records to be able to parse out with confidence which taxon flowered first. Second, with the amount of detailed data cleaning that would need to be done for every taxon, we feel that this type of data collection and analysis would merit its own publication.

To shed some light on the field flowering times of our study taxa, we now include a table of observations from the field and seed collection dates for our study taxa (Appendix 8, Main text line 394). Similar to herbaria records, this is a fairly coarse description of flowering time differences. Personal observations primarily note 'peak' flowering, which could include approximately the middle 50% quantile of the flowering time distribution, where many of our taxa overlap to some degree even if there are divergent flowering onset times. Seed collection dates also have the limitation of being restricted to dates that trips were made to specifically collect – many sister taxa have the same collection date, especially if the fruits of that species are of a type that do not dehisce the seeds quickly. These limitations prevent us from determining the ordering of flowering time for taxa pairs with fairly similar overall flowering time distributions. With these limitations noted, we do recover a confident signal of differences in flowering times for 6 taxa pairs – COSP, NAJP_NAHN, CAGT_CAGA, LADI_LAGL, MNUD_MGUT, and COGR_COSP. These are taxa pairs that have some of the greatest phenological isolation in the greenhouse experiment. For all of these taxa, the order of flowering in the greenhouse is the same as that in the field.

We believe the reviewer misunderstands our discussion on countergradient selection, which is likely due to a lack of clarity on our part. While the countergradient selection operates on the

later-flowering serpentine taxon, the “countergradient” aspect is rooted in three seemingly contradictory results: 1) there is selection for earlier flowering in serpentine soils, 2) there are strong serpentine-mediated plastic shifts to very late flowering in the presumably ancestral-like *nonserpentine* taxon, and 3) the serpentine taxon flowers later in serpentine soil than the nonserpentine taxon in nonserpentine soil, but not as late as the nonserpentine taxon in serpentine soil. The countergradient selection discussion is a speculation on the evolutionary transition to later flowering in the serpentine taxon relative to the nonserpentine taxon when each is in its home soil type. The logic is that the strong serpentine-mediated plastic responses to very late flowering in the nonserpentine taxon represents an ancestral-like condition - initial colonization onto serpentine soil elicits a plastic response to very late flowering, likely due to the nutritional and developmental constraints of serpentine soils. As a new serpentine population adapts to the serpentine substrate (e.g., increasing nutrient uptake/dealing with the heavy metals) it is able to respond to the selection for earlier flowering. This logic explains the pattern in some of the pairs of why, in serpentine soils, the serpentine plants flower earlier than the nonserpentine plants. However, this countergradient pattern (maladaptive plasticity early in the colonization of/establishment on serpentine soils, followed by selection for earlier flowering) is only supported in 6 of the taxa pairs, as noted in the discussion on Lines 412-413.

We do not believe that the countergradient explanation *per se* is due to the greenhouse methodology. For the pairs that show the countergradient pattern, personal observations of the nonserpentine taxa growing in serpentine soils strongly confirm the idea that nutrient limitation causes these plants to flower relatively very late. Most of these plants barely survived— they were small and their leaves often appeared nutrient-stressed (e.g., chlorotic or purple leaves) – and they ultimately produced flowers late in the life cycle. It was clear that the paired serpentine plants did not produce the same symptoms, and that they flowered earlier. The reviewer has a legitimate concern about how disrupting the soil structure (including removing large rocks) would affect the results we see here vs. in the field. We address this below in the context of our finding that serpentine plants flower later than nonserpentine plants when each is in their home soil type (replicating how we would see flowering time divergence in the field). However, this concern shouldn’t affect the countergradient explanation *per se*, as we are really comparing the serpentine and nonserpentine taxa in the serpentine soil. With the presence of rocks, we are confident that the nonserpentine taxa, if they survived, would still flower very late relative to the serpentine plants because of the nutrient limitations. If rockiness/soil structure causes the serpentine taxon to flower earlier in serpentine soils than what we documented in the greenhouse, then the countergradient pattern would be even more exaggerated.

To remedy the confusion around the countergradient selection pattern, we change the language in the discussion to explicitly say that this pattern can rectify the seemingly contradictory results of selection for earlier flowering in serpentine soils and serpentine plants flowering later than nonserpentine plants. Lines 401-404

We believe that the main concern the reviewer brings up here is the discrepancy between the majority of the serpentine literature that suggests serpentine taxa flower *earlier* than their nonserpentine counterparts, and our greenhouse results that show, when each taxon is in its home soil type, serpentine taxa flower *later* than their nonserpentine counterparts. The reviewer brings up the legitimate concern that the disruption of the soil structure and removal of large rocks that

would not fit in the pots we used would change the selective factors (specifically, water availability) acting on flowering time in the greenhouse vs the field. This is a legitimate concern because one of the primary hypotheses for earlier flowering in serpentine populations is that the rocky soil substrates accelerate the onset of the summer drought.

We acknowledge in the manuscript that greenhouse conditions minimize the effects of differences in soil structure between serpentine and nonserpentine soils and discuss the ramifications for our results in the discussion on Lines 385-400. However, the greenhouse transplant experiments still incorporate aspects of soil texture (e.g., fine-grained percentages of sand v. silt v. clay, and rockiness composed of rocks that fit in the pots we used) as well as soil chemistry. As we note in the Discussion (Line 397-398), the serpentine soils we used do have on average more % sand than the nonserpentine soils, which should result in lower water-holding capacities. Unfortunately we did not quantify particle size in the soils that were used in the greenhouse – from personal observation we can attest to there being considerable variation in rockiness in the soil samples that were used in the greenhouse experiment, across both serpentine and nonserpentine populations.

We note however, that disruption in soil structure and a lack of large rocks does not necessarily imply that relative selection, or soil-mediated plasticity, on flowering time in serpentine and nonserpentine populations should be flipped in direction. Moreover, other greenhouse studies of serpentine and nonserpentine taxa in field-collected soils have found patterns in genetic differences in flowering time divergence that parallel those seen in the field (Dittmar and Schemske, 2018). Given that we recover a genetic signal for taxa pairs which have the serpentine taxon flowering later than the nonserpentine taxon, we do not think that selection driven by soil-structure/early-drought would cause genetic differences for earlier flowering in the serpentine taxon. If anything, we may be missing strong plastic effects mediated by soil-structure/early-drought in serpentine soils that could drive serpentine plants to flowering earlier than nonserpentine plants in the field. However, our field observations of flowering times correspond with our greenhouse observations for taxa with marked flowering time shifts. Lastly, not all serpentine habitats are rocky, and plenty of nonserpentine habitats are rocky – thus, it is not surprising that we find a mix of flowering time responses among our taxa.

Dittmar EL, Schemske DW. 2017 The edaphic environment mediates flowering-time differentiation between adjacent populations of *Leptosiphon parviflorus*. *J. Hered.* 109, 90–99. (doi:10.1093/jhered/esx090)

6. ~ Line 382: I believe that the manuscript would benefit from a more detailed section in the Discussion that rectifies the two seemingly contradictory results (later flowering in the greenhouse among serpentine taxa, Fig. 2; and evidence for phenotypic selection for earlier flowering in serpentine soils, Table S6, Fig. S5).

As we note in the prior response, these two results are explained in the context of countergradient selection. We modify the language at the start of that paragraph (Lines 401-404) to explicitly say those two contradictory results can be explained by countergradient selection driving a shift in flowering time.

Minor comments:

INTRODUCTION

Title (and later in the conclusion): Do you have a specific rationale for using the term “speciation spectrum” when the term “speciation continuum” has precedence in the literature? Using the later might help to reduce the amount of jargon within the field and would also alert potential readers to the phylogenetic aspect of your study.

We changed speciation spectrum to speciation continuum throughout.

Lines 38: “that vary in divergence times” might be more appropriate here due to the variability in total phenological RI that you find (i.e., relatively less diverged taxa pairs might still show large phenological difference).

We changed the phrase from “that vary in their progress toward speciation” to “at either ecotypic or species-level divergence”. We are uncomfortable using the phrase “divergence times” since we don’t have exact measures of time. Translating ITS divergence to actual time involves much error. Lines 38-39

Line 72: The phrase “more genetically-based barriers” is unclear here. Perhaps something like “for taxa further along the speciation continuum, we predict that phenological reproductive barriers will be more strongly genetically-based”.

We changed this sentence to “These multiple processes all lead to the prediction that reproductive isolation will be more strongly genetically-based for taxa further along the speciation continuum” Lines 78-80

Line 76: total phenological mismatches = assortative mating; phenological shifts may contribute to RI but are unlikely to result in completely assortative mating.

We changed the clause from “and phenological shifts automatically confer assortative mating” to “phenological shifts automatically increase assortative mating” to acknowledge that phenological shifts don’t necessarily create complete assortative mating. Lines 83-84

Line 84: Give an explicit definition of “countergradient selection” here since it is an important point of the Discussion.

We define countergradient selection in Lines 76-78 of the introduction and in the discussion in Lines 413-415.

Line 87: perhaps... FTs promotes “the long-term evolution of RI, on timescales important to speciation”.

Thank you for the suggestion – we incorporated this in the manuscript. Line 91-92

Line 94: Perhaps expound on specific examples as in [33,34] for readers less familiar with serpentine.

We changed the sentence to: “Serpentine soils are harsh, often rocky substrates, characterized by low Ca:Mg ratios, low nutrients and high heavy metals, and they impose strong divergent selection across steep ecological gradients [33,34]” Line 96-98

Line 94: “at least” 39 families? I don’t think Anacker et al. 2011 were 100% comprehensive, but relied on the existence of sequence data when available to reconstruct trees.

You are correct about the dataset in Anacker et al ’11, and we added the “at least” as suggested here. Line 99

Lines 97-99: Is it important to mention the possibility of peripheral-isolate speciation here? Do we necessarily expect in situ adaptive divergence (for many readers I think this will insinuate speciation with gene flow), to be more prevalent than micro-scale allopatry, facilitated by California’s heterogeneous environment? The evolution of phenological RI seems very likely under a budding speciation scenario where derivative lineages occur at the periphery of progenitor species, perhaps in different edaphic contexts.

I think you are correct with this interpretation - we actually expect micro-scale habitat heterogeneity, facilitating peripheral-isolate speciation, to be far more prevalent than ‘in-situ’ (sympatric) adaptive divergence.

Using ‘in situ’ divergence was meant to distinguish between two hypothetical evolutionary routes to endemism: neo- and paleoendemism. The neoendemic pathway is often described as a peripheral-isolate speciation process (i.e., colonization and adaptation to a new habitat, accompanied by/followed with the evolution of reproductive isolation). In contrast, the paleoendemic pathway involves a tolerator species that loses all of its nonserpentine populations of over time (e.g., nonserpentine habitats become more competitive and only populations already on serpentine, or that migrate to serpentine, persist). Neoendemism is thought to be the predominant route to endemism in the annual CA serpentine flora.

We replace “and are thought to arise through in situ adaptive divergence and speciation” with “are thought to arise through budding, or peripheral-isolate, speciation” Line 103

Lines 99-101: Is there any evidence that soil chemistry is related to flowering time? I understand that you are somewhat constrained in your interpretations of how soil chemistry and soil texture might be the casual mechanisms driving differences in flowering times in the field vs. greenhouse. Some consistency between the intro and conclusion would provide a more coherent argument – either chemistry-chemistry in both, or texture-texture in both, with a discussion of how soil transplants in the GH often disrupt the structure of the soil.

We modified these ideas in the intro to describe how the majority of earlier flowering is attributed to drought avoidance in rocky soils, but we also add that the life history expectation of

flowering in nutrient stressed habitats is to delayed flowering (Lines 104-107). We think it is important to include both texture and chemistry in the introduction and discussion, as our greenhouse experiment does not fully negate texture, but we also include a discussion of how greenhouse soil transplants disrupt soil structure and remove large rocks (Lines 385-400).

Line 104: Something like “17 taxon pairs representing the independent evolution of edaphic divergence” might be more clear here.

We make this sentence more clear with “we examine FT divergence across 17 taxon pairs that represent independent adaptation to serpentine soil leading to either a serpentine tolerator or a serpentine endemic” Line 110-111

Lines 106-109: Why do you hypothesize this? Is this purely a time since divergence argument, or a permeability of barriers argument? Do you expect plasticity to facilitate initial phenotypic (phenological) differences, and then subsequent adaptation to the edaphic environment to result in elevated phenological divergence?

This comment is similar to reviewer #3’s comment, and we motivate this hypothesis more fully in the second paragraph of the introduction. We hypothesize that taxa further down the speciation continuum will have more genetically-based barriers because permeable plastic barriers should limit progress to speciation. We also acknowledge that this prediction may be due to more diverged taxa having had more time to accumulate genetically based barriers, or to initially plastic barriers being made permanent by canalizing selection. Line 67-80

Lines 114: We estimate divergence times of sister pairs and serpentine-nonserpentine population pairs using ITS sequence divergence. Without mentioning this here, readers might be left to wonder how you were able to ask the question in Lines 121-123.

We do not add this sentence here, as we did not use ITS sequence divergence to quantify divergence times *per se* (see response to Major Comment #4).

Lines 121-123: This approach/hypothesis (and potentially the statement in Lines 106-109) supposes that sister “species” (serpentine endemics – close relatives) are in fact more diverged than tolerator serpentine-nonserpentine populations. This brings up the question of taxonomic inconsistencies across groups, as well as that of asymmetrical rates of molecular evolution and/or sequence divergence. Both of which likely warrant some discussion later in the manuscript.

Please see responses to Major Comment #4.

METHODS

Lines 146-150. This is a nice approach. I think it is definitely worth mentioning here in the main text that you largely estimated ITS divergence using the very same populations used in the experiments! This often isn’t the case, and leaves readers wondering if specific isolated populations might show much different levels of genetic divergence compared to other populations from which tissue was sampled. Overall, your approach allays some of the concerns

associated with taxonomic inconsistencies (i.e., a “species” in one group is no different than a divergent population pair in another group); however, I still wonder how sensitive your results are to heterogeneous rates of molecular evolution observed in different lineages. As you know, Kay et al. 2006 found substantial variation in ITS substitution rates among herbaceous annuals and perennials from California. A comparison of ITS substitution rates within a single lineage would provide a relatively reliable estimate of divergence time among taxon pairs. I realize that comparing across lineages is logistically the only practical approach for this study, but I think it might be worth mentioning this limitation as a potential caveat in the later results/conclusions pertaining to endemic-sister and serpentine-nonserpentine comparisons.

Additionally, I think it should be mentioned that the mean values presented here are strongly driven by substantial divergence in the single *Collinsia greenei* – *C. sparsiflora* species pair (21 substitutions). The big point being – how much of one of your major results (i.e., high phenological isolation is associated with initial ecotypic or population-level divergence) is an artifact of “species” (mean ITS divergence = 0.56) and populations (mean ITS divergence without the *Collinsia* pair = 2.29) having generally similar rates of ITS divergence? Maybe the distinction between species and populations is arbitrary here? It appears that 0.56 vs. 2.29 substitutions is statistically significant here, but it might merit additional discussion, and/or also be worth indicating in approximate years what this difference in substitutions represents (using values from Kay et al. 2006).

As noted above, we stay away from using ITS sequence divergence as a quantitative measure of divergence time, but rather to support the taxonomic circumscriptions that we use to define within- vs. between-species comparisons. We clarify our intention in the methods section on Line 155-156.

Lines 158-160: Please provide more details on soil collection. 15L from a single hole? 15L sampled from underneath the same 30-40 maternal plants? What was the justification to remove larger (>3.5cm rocks)? Were soils homogenized in buckets before being brought to the greenhouse, or was the soil structure/horizon maintained in individual pots? You circle back in the conclusions to the idea that you cannot attribute FT differences in your experiment to soil structure, but only to soil chemistry. Certainly this may be the most stringent interpretation of your results, but I wanted a little more detail here to decide for myself!

We added to the methods section that the 15L of soil were collected from 5-6 locations within each population. We specified that we discarded rocks that would not fit into the Conetainer pots we used (pot diameter = 3.8 cm). We added a sentence that we homogenized soil from each population before using it in the experiment. Lines 161-168

See Kelso et al. 2003, Geobotany of the Niobrara Chalk Barrens in Colorado: a study of edaphic endemism, in *Western North American Naturalist* for a good discussion of the importance of soil structure in shaping endemism. My general worry with soil transplant experiments in the greenhouse, is that the structure of the soil is disturbed during collection, such that many of the factors shaping its putative selective pressures are erased or altered compared to field conditions. That said, do you have any data (i.e., water holding capacity of serpentine soils in the cones vs. nonserpentine soils) to speak to this? I don't want to see you totally discount the

effect of soil texture on your results, especially since you have data from the 2019 paper that may speak here.

This is a great reference, and we acknowledge that the selective factors stemming from the soil structure are altered in a greenhouse experiment in Lines 385-400. Unfortunately, we do not have data for water holding capacity of the different soils we used. We also do not believe that the soil texture effects are negligible. There are still differences in the soils in the composition of the fine soil fraction (% sand, silt and clay) that were quantified in our 2019 paper. Although we do not have this quantified, there was certainly variation in rockiness in the soils that were used in the greenhouse – i.e., some soils were largely composed of rocks < 3.5 cm – both in serpentine and nonserpentine soils.

On a related note, serpentine soils themselves often possess high amounts of silt and clay, leading to high water holding capacity. Thus, the assumption that rocky, serpentine sites in the field experience drier conditions (and thus selection for earlier flowering) might result more from site effects (i.e., lack of substantial shrub and tree vegetation, leading to a lack of shade and full sun exposure), as opposed to inherent soil effects. The offshoot is that in the greenhouse, which presumably has a consistent degree of shading/sun across its extent, potting soil can actually dry out faster than serpentine soils when in containers, specifically once large rocks are removed from the serpentine mix. Did you assess soil moisture in the GH pots throughout the experiment? Thus, the countergradient variation observed may result mostly for methodological reasons.

We did not monitor soil moisture in the GH pots. It is true that serpentine soils themselves can have high levels of silt and clay, although we found in our 2019 paper that the serpentine soils had high % sand content (and lower % silt and % clay content) than the nonserpentine soils. That said, we do not have a quantification of the % of soil composed by coarse vs fine fraction. This is something that, from personal observation, did vary in both the serpentine and nonserpentine soils but we do not know if it was greater in one soil type.

We do not think the countergradient pattern results from mostly methodological reasons (see response to Major comment # 5).

For the main finding that the serpentine taxon flowers later than its paired nonserpentine taxon in their home soil types, the disruption of soil structure could have affected our results (see response to Major comment #5). However, the taxa pairs for which we have confident observations from the field on flowering time differences (new Appendix 8) all have similar timing patterns (i.e., which taxon flowered first) as they did in the greenhouse.

Lines 177-180: How did you deal with this when calculating overall flowering windows, and thus overall phenological RI?

We used the data as collected for the overall flowering windows, and did not make any adjustments for the lack of self-pollination. Luckily, the vast majority of the pairs had the same mating system – either both setting self-fruit or not setting any fruit. In the case of taxa that did not set self-fruit, their flowering time windows are undoubtedly longer than if they had been

pollinated. Given that the start of the flowering window would not change regardless of whether the plants were pollinated or not and that those plants likely produced more flowers than they would have if pollinated, we believe that the flowering windows of plants that didn't self-pollinate would just be stretch out – this should have the effect of there being more overlap than there would otherwise be if they had been pollinated, and should thus underestimate RI. Given that mating systems were evenly spread across endemic and tolerator pairs, we do not believe this biases our results between endemic and tolerator pairs.

Lines 186-190: Is a t-test the most appropriate test here? I'm wondering if there was substantial variation among maternal families, and/or whether a regression/ANOVA approach that included a random effect for maternal family might be more appropriate? If not, a t-test seems great.

The goal of this analysis was to test for differences, within a pair, between two treatment: the nonserpentine (NS) taxon in the NS soil, and the serpentine (S) taxon in the S soil. We think the t-test is the appropriate test here. Within a given treatment combination (e.g., S taxon in S soil) each maternal family is represented with 1 individual, so it would not make sense to parse out variation in flowering time in that treatment due to maternal family.

Lines 192: Is it worth looking at a chemical PCA, or serpentine harshness index independently from a soil texture metric? I know you're balancing a lot here, but the % variance explained between the chemical aspects of the soil and the texture aspects of the soil, might buy you a little more ability to make stronger claims in the conclusion, and/or not have to shy away from asserting an effect for soil texture.

We explored this idea by rerunning the soil PCA with either chemistry variables only or texture variables only, and then reran the PGLS analyses with soil distances based on the new PC scores. The chemistry-only analysis (top panel below) was marginally significant and explained 23% of the variance (similarly to the chemistry+texture analysis that we have in the manuscript, which was marginally significant and explained 23% of the variance; middle panel below). The texture-only analysis (bottom panel) was not significant and explained 0.1% of the variance. Thus, our measures of soil texture do not have a strong predictive relationship to flowering time. We would prefer to leave these extra analyses out of the manuscript because we don't feel they add anything. However, if the editor prefers, we could add them to the supplementary materials.

Lines 201-203: Is the “phylogenetic relatedness” here simply ITS sequence divergence? If so, perhaps be explicit, and ignore many of my earlier comments related to the topic!!

Phylogenetic relatedness is based on the ITS sequences. As in our 2019 paper, we build a phylogenetic tree from our ITS sequences, and use the phylogenetic variance-covariance matrix to parameterize a random-effect in the Bayesian models. The random effect controls for the fact that we expect trait values to be more similar for closely related taxa.

Lines 219-222: Please provide more rationale for these expectations. See my thoughts in the “Major Comments” section above.

We added more clarity to the beginning of this methods section to explain our rationale for how we parse out the genetic vs plastic contributions to flowering time divergence. Lines 223-231

Lines 238-241: How did you rectify flowering time variation (lines 179-180) among nonpollinated and non-selfing GH plants with those that selfed, when calculating phenological RI?

See comment above (starts with “Lines 177-180”)

Lines 246: “did not”

Corrected

Figure 1: Excellent conceptual figure! For the text pertaining to Figure 1C, I think readers need more detailed explanation for how/why the treatment contrasts pertain to plastic (1 vs. 2) and genetic (2 vs. 3) contributions.

Because we are at the word limit, we do not add more text in the figure legend, which goes to the word count. We added in more detail about our logic for using these contrasts in the methods on Line 233-231. We also do not want to add more text to the figure itself, as this was a major criticism of the first rendition of the figure.

Also, why isn't the 1 vs. 4 contrast given more consideration? Despite typically high mortality of nonserpentine taxa on serpentine soils, there must be some cases in which there was survival. This scenario likely represents the colonization of serpentine substrates by ancestral nonserpentine lineages, and might be indicative of early selection pressures associated with soil shifts. Despite 30-40 maternal families per population, was there just insufficient population level sampling to make this a meaningful component of the study? This is certainly outside the scope of your study, but it would be really interesting to do a similar experiment with say 20-30 tolerant but nonserpentine populations that occurred in habitats that varied in their degree of “serpentine-ness”.

We do not focus to the 1 vs 4 contrast because, for the majority of the pairs, there was very low survivorship. We will be documenting these fitness differences in a subsequent paper. Because Reviewer #3 asked to include the full flowering time distributions of every pair in each of the ecological contexts in the supplement (e.g., when taxa are in their home soils, or the common nonserpentine soil), we also include a graph for the flowering time distributions in serpentine soil.

For nonserpentine taxa that do survive in serpentine soil, we document serpentine-soil-mediated plasticity in the NS taxa as a proxy for what early responses to serpentine upon colonization may have been. This dataset forms, in part, our discussion about countergradient selection. Lines 262-274; 337-346.

Figure 1: 1D please define circles and triangles. Why is “The difference between the two ecological contexts reflects the permanence of phenological isolation” true?

The circles and triangles reflect the nonserpentine and serpentine taxa, respectively, and are defined in Part A of the figure. The difference in phenological isolation between the two ecological context – when taxa are either in their home soils or a common nonserpentine soil – indicates whether the magnitude of phenological isolation changes in different environmental contexts, presumably due to plasticity. If phenological isolation is largely genetically based, we expect little change when we measure RI b/t sister taxa in their home environment, and RI b/t sister taxa in a common nonserpentine environment. Again, we do not focus on RI in a

serpentine environment, because the majority of nonserpentine taxa have low survival. However, if the flowering time shifts of serpentine plants are largely serpentine-soil-mediated plastic changes, then we expect there to be a reduced in RI when measured in a common nonserpentine soil. We believe that this logic is conveyed in the text within panel 1D, and it is also described on the main text in lines 86-89; 133-135

RESULTS

Figure 2: Caption to include, “positive values reflect...later flowering in serpentine taxa”.

We initially had this information in the body of the figure, but were urged to remove the text from the figures by the first round of reviewers. We believe that the second line of the y-axis and the first line of the caption are enough information for the readers to infer the biological interpretation of positive v. negative values.

Line 281: with shifts in the onset of flowering? Perhaps remind reader of your definition of FT at the start of this section.

We changed the sentence to “...with shifts in FT (the onset of flowering).” Line 291

Lines 266: ...with an average shift of xx days later (Fig. 2).

We added this information to the end of the sentence. Lines 296

Lines 287: Is your composite multi-variate soil metric the most important thing to be examining here? How much variation does soil texture alone explain? What about texture vs. chemistry?

Please see response above to a similar comment (starts with “Lines 192”)

Lines 297: This is a very interesting result! I’ll be curious to hear your interpretation in the Discussion, but it seems to me that give the mean shift of 19.4 days, and only the 0.58 differences between endemics-sisters vs. serpentine-nonserpentine pairs, that most of the shift is happening early during speciation! (again, assuming that we are confident in divergence time estimates across lineages).

Figure 3: I’m wondering if it’s worth adding a third panel that shows the mean +se of genetic and plastic for both the tolerator and endemic pairs? Or adding the averages in the text (as below)?

We added these averages + se to the text (as in the comment below) Lines 315-319

Lines 302-302: genetic effect (average xx%).....plastic effect (average yy%)

See response to prior comment.

Lines 311-312: Cool! Again, suggesting that RI associated with phenology might evolve early, but then be constrained from increasing beyond some bound. Certainly RI is mathematically constrained by a value of 1, but I'd but curious to hear more in the Discussion as to what constraints might limit the increased evolution of phenological differences among species vs. ecotypes?

We are uncertain why phenological RI is not stronger for endemic than tolerator pairs, and don't feel it is worth the space to speculate. It may simply be that flowering time is constrained by the relatively short growing season in this Mediterranean climate ecosystem, or it may be that it relates quantitatively to divergence time (which we don't know with precision here), as mentioned in Line 426-428.

Lines 316: and perhaps as a Supplementary Figure – is there a relationship between divergence time and RI phenology (or at least within either class of pairs)?

Because of the reasons listed in our response to Major Comment #4, we do not attempt to quantitatively determine divergence times from the ITS sequences nor use those approximates in quantitative analyses. Our ITS sequence data is used to primarily support the taxonomic circumscriptions of our pairs.

Figure 4: Again, if survival sample size allows, I'd love to see a paired figure that also examines non-serp taxa growing in serp soil.

As noted above, we have included a supplementary figure that shows the flowering time distributions for all of the serpentine and nonserpentine taxa in serpentine soils. However, because the majority of pairs have very low survivorship of nonserpentine individuals, we do not formally quantify phenological isolation and compare it between the endemic and tolerator pairs.

Lines 341: I wonder if transferring Fig. 4 to the supplement (to me the figure doesn't convey much additional information than is present in the text), and moving Figure S5 to the main text would be worthwhile? This could also potentially give the Discussion a nice angle, in that this result tends to confirm our expectation for drought avoidance on serpentine, and would provide a contrast to the countergradient trend.

We take your advice to move Fig. 4 to the supplement, but we leave former Figure S5 (now S9). We believe the phenotypic selection analyses that show selection for earlier flowering (Figure S9), combined with the reaction norms in Figure S8, support our countergradient selection hypothesis.

DISCUSSION:

Lines 345: Independent “instances”? Replicates here might suggest multiple nonserpentine to serpentine transitions within a single lineage.

We choose to keep replicates here, as these taxa pairs arguably represent evolutionary independent events of edaphic divergence.

Lines 349-350: What is the relationship between “chemically harsh” soils and flowering time? Is there much past work here to inform an expectation? It seems to me that the real or putative “dryness” of serpentine soils lead to an expectation for earlier flowering, but may not result from their chemical compositions. I suggest adding some background on this potential relationship in the Intro, or at least also mentioning structural/textural characteristics of serpentine here (even if you cannot definitively pinpoint the casual mechanism from your data/experimental design).

We both add more background in the intro (Lines 104-108) and change the phrasing here to “chemically and physically harsh” in order to incorporate both the chemical and physical components of serpentine.

Lines 361-362: Isn't this the other way around? Colonization of new habitats may facilitate FT shifts and ecotypic divergence. Why would genetic variation within singular habitats be maintained if FT was already at an adaptive optimum?

This paragraph discusses a hypothesis by Levin (2010) wherein colonization of a marginal habitat results in initially plastic flowering time shifts, and that these flowering time shifts acts as a partial reproductive barrier to maladaptive gene flow from source populations, allowing the marginal populations to adapt to their new habitat. We view our results through the lens of this hypothesis in this paragraph. We changed the first sentence of this paragraph to improve clarity (Lines 375-376).

Levin DA. 2009 Flowering-time plasticity facilitates niche shifts in adjacent populations. *New Phytol.* **183**, 661–666. (doi:10.1111/j.1469-8137.2009.02889.x)

Lines 372-375: How can we rectify this with the results from Table S6 and Figure S5?

The countergradient explanation one way to rectify the fact that we see selection for earlier flowering in serpentine soils, and yet serpentine plants flower later than nonserpentine plants. Please refer to the responses to Major Comment #5.

Lines 378-379: Did you quantify gravimetric soil moisture in greenhouse cones? Given the lack of rocks in the cones (and absence of full sun in the GH), it's possible that they held more water under greenhouse conditions? Overall, I think this paragraph would benefit for ecological/methodological explanations for how experimental conditions might be different from natural conditions.

We did not quantify soil moisture in the greenhouse cones, nor water holding capacity of soil in the greenhouse cones. It is possible that the greenhouse conditions experienced more water than

the field conditions, given the disruption of the soil structure and the inability to precisely mimic field rainfall patterns. Although we didn't measure water holding capacity of the soils in the greenhouse cones, they undoubtedly differ among the soil samples used (from personal observations of rockiness). We expanded this paragraph in the discussion to include how the conditions in the field may be different than those in the greenhouse. Also, see responses to Major Comment #5

Lines 389-394: Very interesting idea!

Lines 403-405: Have you regressed the strength of RI phenology against ITS sequence divergence? Potentially just within ecotypes?

Please note responses above (for Major comment #4 and Specific comment starting with "Line 316").

Lines 408-409: re: soil chemistry and its potential effects on FT, are there any cases in the literature that document FT shifts in response to soil chemistry.

We are unaware of any studies that isolate the effects of serpentine chemistry *per se* on flowering time, and we do not focus on the effects of chemistry only as we cannot detangle them for differences in the texture that we do capture with soil transplants.

Lines 419: It seems to me that one of the largest contributions of this work is that RI associated with flowering times differences is substantial early in the divergence process. I think you should make a bigger deal of this result, and fit it into the broader plant speciation literature a bit more before the conclusions. Also, why do flowering time differences evolve substantially at first, but then less so afterwards (i.e., what is the constraint or conservatism at play here)? Is this phenomenon only restricted to edaphic speciation, or how might it be associated with other habitat shifts, long distance dispersal, etc.?

We place our phenological isolation results in the broader context of phenological isolation in other systems, and generally in the context of the importance of ecological divergence in driving the early stages of plant speciation, in the second paragraph of the discussion (Lines 366-365).

Please refer to response to comment starting with "Lines 311-312" regarding constraints on phenological isolation.

Lines 423: speciation "continuum"?

Changed.